# Influence of Quenching and Subsequent Artificial Aging on Tensile Strength of Laser-Welded Joints of Al–Cu–Li Alloy

**Alexandr Malikov [1,\*], Evgeniy Karpov [1,2], Konstantin Kuper [1,3] and Alexandr Shmakov [1,4]**

1    Khristianovich Institute of Theoretical and Applied Mechanics, SB RAS, 4/1 Institutskaya Str., 630090 Novosibirsk, Russia; evkarpov@mail.ru (E.K.); k.e.kuper@inp.nsk.su (K.K.); highres@mail.ru (A.S.)
2    Lavrentyev Institute of Hydrodynamics, SB RAS, Lavrentyev Ave. 15, 630090 Novosibirsk, Russia
3    Budker Institute of Nuclear Physics, SB RAS, Lavrentyev Ave. 11, 630090 Novosibirsk, Russia
4    Boreskov Institute of Catalysis, SB RAS, Lavrentyev Ave. 5, 630090 Novosibirsk, Russia
\*    Correspondence: smalik@ngs.ru

**Abstract:** The research aim was to optimize post-weld heat-treatment (PWHT) modes for a laser-welded joint of the Al–Cu–Li alloy and improve their respective strength properties. As a result, the ultimate tensile strength, yield point, and elongation of the joint were enhanced up to 95%, 94%, and 38%, respectively, of those inherent in the base metal. Before and after PWHT, both microstructures and phase compositions have been examined by optical and scanning electron microscopy, as well as synchrotron X-ray diffractometry. In the as-welded metal, the $\alpha$-Al and $T_1(Al_2CuLi)$ phases were found, along with the $\theta'(Al_2Cu)$ and $S'(Al_2CuMg)$ phases localized at the grain boundaries, significantly reducing the mechanical properties of the joint. Upon quenching, the agglomerates dissolved at the grain boundaries, the solid solution was homogenized, and both Guinier–Preston zones and precipitates of the intermediate metastable $\theta''$ phase were formed. After subsequent optimal artificial aging, the (predominant) hardening $\theta'$ and (partial) $T_1(Al_2CuLi)$ phases were observed in the weld metal, which contributed to the improvement of the strength properties of the joint.

**Keywords:** laser welding; aluminum-lithium alloy; scanning electron microscopy; synchrotron radiation; X-ray diffractometry; mechanical properties

## 1. Introduction

Aluminum–lithium alloys are widely used in the aerospace industry due to their unique properties, combining low density with high-strength characteristics and an elastic modulus. To date, the third generation of the heat-treatable wrought grades of the Al–Cu–Li–X systems have been designed (X = Mg, Zn, Mn, Zr, Sc, Ag) [1–6].

It is well-known that high mechanical properties of aluminum–lithium alloys can be achieved by appropriate heat treatment (HT) or thermomechanical processing. Detailed studies of various HT procedures (hardening, artificial aging, annealing) are being carried out [1–7]. For example, artificial aging leads to the precipitation of the hardening $T_1(Al_2CuLi)$, $\delta'(Al_3Li)$, $\theta'(Al_2Cu)$ and $S'(Al_2CuMg)$ phases from solid solutions. As a result, both phase composition and mechanical properties are changed as well.

The strength–ductility paradox is a common dilemma in the development of high-strength copper-based alloys because of their grain refinement upon critical supercooling [8]. This phenomenon is a typical example of nonequilibrium structures formed during solidification of a highly super-cooled alloy due to its recrystallization [9].

In order to replace the riveting method for joining parts in the manufacturing of aircrafts and, consequently, reduce their weight; laser-welding procedures are being actively developed for aluminum–lithium alloys [10–15]. In particular, Ning et al. have investigated laser welding of the 2A97 grade (Al-3.9Cu-1.4Li) with the ER2319 filler wire (Al-5.6Cu) [16]. As a result, they have suggested that the $T_1(Al_2CuLi)$ phase has been formed in the weld

metal at the grain boundaries, while the $\theta'(Al_2Cu)$ one has been observed inside them. The ultimate tensile strength of the joint has been 59.8% from that inherent in the base metal. For laser-welded joints of the same 2A97 alloy, Fu et al. have reported [17] that the eutectic may contain $T_1$, $T_2$, $S'$, or $\theta'$ phases in the weld metal along the grain boundaries, but concentrations of the $\delta'$, $\beta'$, $\theta'$, and $T_1$ phases may be reduced inside them. The ultimate tensile strength of the welded joint has been 83.4% of that for the base metal. Laser welding of the 2060 alloy (Al-3.95Cu-0.75Li) with various filler wires has been investigated by Zhang et al. [18,19]. The use of the 5087 wire has enabled to form the $T_2Al_6Cu(Li, Mg)_3$ and $\delta'(Al_3Li)$ phases in the weld metal, while the LiAlSi, $\theta'(Al_2Cu)$, and $Mg_2Si$ phases have been found after welding with the $AlSi_{12}$ filler. In the last case, the ultimate tensile strength of the welded joint has been 80% of that for the base metal. Liu et al. have investigated laser welding of the 2060 alloy (Al-3.95Cu-0.75Li) with the ER 2319 filler wire [20]. As a result, secondary-phase particles (SPPs) have been formed in the weld metal (for example, $\theta'(Al_2Cu)$) with very small numbers of other inclusions, such as $Al_{11}Cu_5Mn_3$ and $Al_7Cu_2Fe$. Zhang et al. have examined the microstructure and corrosion resistance of laser-welded joints of the 2A97 alloy [21]. In the weld metal, it has been found that the $T_1(Al_2CuLi)$, $T_B(Al_7Cu_4Li)$, and $R(Al_5CuLi_3)$ phases, distributed inhomogeneously between dendrites. Han et al. have applied double-side laser welding with the CW3 (Al-6.2Cu-5.4Si) and AA4047 (Al-12Si) filler wires for manufacturing dissimilar T-joints of the 2060 (Al-3.9Cu-0.8Li) and 2099 (Al-2.95Cu-1.87Li) alloys [22]. In the weld metal, both T (AlLiSi) and $T_2$ ($Al_6CuLi_3$) phases have been observed. After using the AA4047 wire, the main hardening phase has been the T one, while it has been the $T_2$ phase in the case of CW3. When studying the weld metal of the 2195 alloy (Al-4.05Cu-0.96Li), Wang et al. have found the $\delta'(Al_3Li)$ and $\theta'(Al_2Cu)$ phases [23]. In [24], Faraji et al. have investigated the possibility of hybrid laser welding of the 2198 alloy (Al-3.5Cu-1.1Li). As a result, they have reported the optimal energy parameters.

Xu et al. have investigated the effect of using the ER2319CT and ER4047 filler wires on some characteristics of laser-welded joints of the 2195-T8 alloy (Al-Li) [25]. Neither equiaxed grains nor columnar dendrites have been found in the weld metal after using the ER2319CT wire. Both AlLiSi and $Al_2Cu$ phases have been identified as hardening after welding with ER4047, while the $\theta'(Al_2Cu)$, $T_1(Al_2CuLi)$, and $T_2(Al_6CuLi_3)$ have been hardening in the ER2319CT case. Compared to autogenous laser welding, a greater amount of the hardening phases improves the mechanical properties of the joints after using the filler wires.

It should also be noted another effective method of joining Al–Cu–Li alloys, namely friction stir welding [26,27].

After laser welding, the ultimate tensile strength of the joints is about 0.60–0.85 of that characteristic of the base metal. In these cases, the decrease in the mechanical properties is associated with a sharp change in the microstructure and phase composition of the weld metal, caused by redistribution of alloying elements in the solid solution upon melting and subsequent solidification. In particular, copper precipitates at the grain boundaries.

It can be concluded based on the above that the phase composition of a weld metal depends on the ratio of alloying elements (copper, lithium, and magnesium) in a base metal, as well as on the use of a filler. In addition, distributions of hardening phases are different in a solid solution and at the grain boundaries. In general, their presence in the weld metal does not reveal any correlations with changes in the mechanical properties, primarily the strength reduction. Any technique for controlling the microstructure and phase composition in the weld metal has not been developed so far.

It is well-known that thermomechanical processing improves the mechanical properties of similar welded joints of heat-treatable aluminum–lithium alloys (and dissimilar ones in some cases) [28–31].

This study, which is a continuation of [32], is devoted to changes in the microstructure and phase composition of the weld metal of the Al–3.9Cu–0.3Mg–1.2Li alloy through its post-weld HT (PWHT). The PWHT procedure has included quenching and subsequent

artificial aging, affecting the mechanical properties of the joint. The principal feature of the study is that PWHT has been carried out for welded samples, which are significantly inhomogeneous in terms of their microstructure and phase compositions, as well as the mechanical properties. In such cases, conventional HT procedures, used for manufacturing rolled Al–Cu–Li alloys, are not applicable. Therefore, it has been necessary to optimize PWHT modes, since the phase composition of the base metal should not be changed, but hardening phases should be formed in the weld metal. Before and after PWHT, both microstructures and phase compositions have been examined by optical microscopy (OM) and scanning electron microscopy (SEM), X-ray diffraction (XRD), and differential thermal analysis (DTA). For the first time, synchrotron radiation has been used to investigate the spatial distribution of phases in the weld metal.

## 2. Materials and Methods

The studies used high-strength corrosion-resistant alloy V-1469T1 (Al–Cu–Mg–Li system) [3]. Two plates with a thickness of 1.5 mm from the V-1469 alloy designed by the All-Russian Scientific Research Institute of Aviation Materials [3,33,34] were joined using a $CO_2$ laser with a maximum power of 8 kW, which is a part of the "Siberia" installation developed by the Khristianovich Institute of Theoretical and Applied Mechanics of the Siberian Branch of the Russian Academy of Sciences. As the main metal for welding, an alloy with an initial dispersion strengthening after aging treatment was used.

Figure 1 shows a scheme of the welding process and subsequent preparation of samples for their examinations, including tensile tests, both OM and SEM metallographic studies, and the phase analysis by synchrotron radiation at a "megascience" facility.

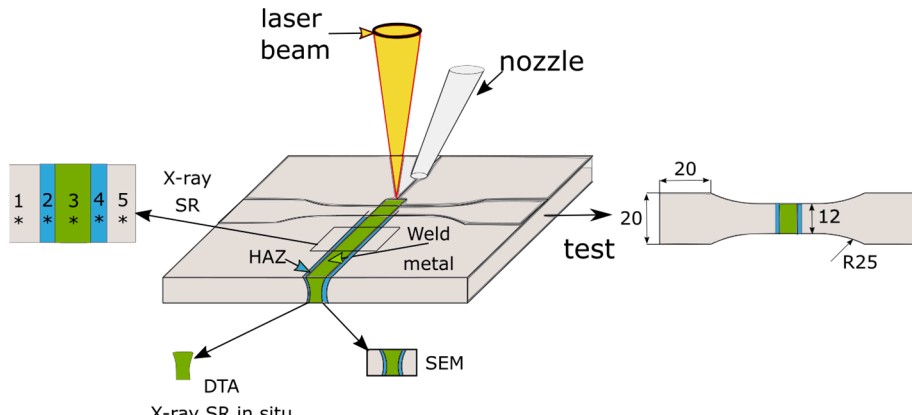

**Figure 1.** The scheme of the welding process and subsequent preparation of the samples for the metallographic studies. * Measuring points.

The procedure of the sample preparation for welding was described in [35].

A laser beam was focused on the plate surfaces using ZnSe lens with a focal distance of 254 mm. On the lens, the beam diameter was 25 mm. In the waist, it was estimated to be 250 μm by the addition of those for diffraction and dissipation spots, caused by spherical aberration.

Both weld zone and its root were shielded with helium at a flow rate of 5 L/min. From the side of the weld pool, the gas was fed through a nozzle tilted at an angle of 45° to the plate surfaces.

After welding, the "dog-bone" samples (Figure 1) were fabricated by milling for the tensile tests according to ISO 4136.

The applied PWHT procedures included quenching in water after exposure in a muffle furnace at temperatures of 500, 530, and 560 °C for 30 min. The artificial aging temperature range was within 160–180 °C, and its duration was 24–40 h.

The heating rate was 5 °C/min in all cases.

Using synchrotron radiation, X-ray phase analysis was carried out in ex situ and in situ modes. The ex situ model was applied to determine phase compositions of the base and weld metals before and after PWHT. The in situ mode was used to assess the temperature-depending changes in the phase composition of the weld metal upon its heating. The studies were carried out with the synchrotron beam of the VEPP-3 storage ring at the Siberian Center for Synchrotron and Terahertz Radiation of the Institute of Nuclear Physics SB RAS (Novosibirsk, Russia). Diffraction from the sample was recorded with an XY detector based on the Mar345 memory screens. When reflections of individual phases were represented in figures, the resulting diffraction rings were integrated over the radius and recalculated for $\lambda = 1.5406$ Å (CuK$\alpha$ radiation). The pixel size was $100 \times 100$ μm. The in situ studies were performed at the "High-Precision Diffractometry II" station of the sixth channel at the same storage ring. During the experiment, the sample was heated from a room temperature up to 600 °C at a rate of 5 °C/min. The X-ray analysis was carried out at a wavelength of 0.164 nm. Then, the resulting diffraction patterns were recalculated into diffraction angles corresponding to the characteristic radiation of an X-ray tube with a copper anode of 0.15406 nm (CuK$\alpha$ radiation).

Additionally, DTA, thermogravimetric analysis (TGA), and differential scanning calorimetry (DSC) were performed with an SDT Q600 TGA/DSC/DTA synchronous thermal analyzer (Netzsch, Selb, Germany) to assess changes occurring in the welded joint upon heating. The recorded variations of the sample weight and detected processes were accompanied by the release or absorption of heat. The sample was heated in the temperature range of 30–600 °C at a rate of 1 °C/min in an inert argon atmosphere (a flow rate of 20 mL/min) in order to exclude the influence of oxidative reactions on the results of the study.

The microstructure and chemical composition of the base and weld metals were examined by SEM and energy-dispersive X-ray spectroscopy. For this purpose, a MERLIN Compact Microscope (Carl Zeiss, Jena, Germany) was employed. Preliminary, specimens had been cut out and prepared with an automatic polishing machine (Presi, Eybens, France). The polished specimens had been etched with Keller's reagent for 30 s.

## 3. Results

### 3.1. General Patterns

The laser-welding-energy parameters were optimized according to the criterion of the absence of external discontinuities, such as cracks, incomplete weld metal, undercuts, shrinkage voids, surface pores, etc. Table 1 presents the optimal energy parameters for obtaining sound welds ~1.8-mm thick.

**Table 1.** The optimal energy parameters for laser welding.

| $W$, kW | $V$, m/min | $\Delta f$, mm | $P$, J/mm | $E$, J/mm$^3$ |
|---------|-----------|----------------|-----------|----------------|
| 3.5 | 4 | −3 | 20.99 | 49.5 |

$W$ is the laser radiation power; $V$ is the welding speed; $\Delta f$ is the focusing depth; $P = W/V$ is the heat input; $E = W/Vth$ is the energy per unit volume of the molten material with the $t$ thickness and the $h$ width.

Figure 2 shows a general view of the cross-section of the welded joint, as well as the microstructure of the weld metal (WM), the heat-affected zone (HAZ), and the base metal (BM).

Equiaxed dendritic grains, formed upon solidification of a weld pool, were found in the weld metal, while long and oriented columnar crystal structures were observed in the HAZ. In turn, the HAZ included the partially molten zone (PMZ) and the fine equiaxed zone (FQZ). The PMZ consisted of spherical equiaxed grains 10–15 μm in size. Within the PMZ, a large number of fine equiaxed grains with close sphericity had been formed, since they had not had enough time and space to grow their branches and contact each other. It is currently believed that the presence of the $\beta'$(Al$_3$Zr) and Al$_3$(Li$_x$,Zr$_{1-x}$) phases is the mechanism for the formation of such equiaxed grains in the base metal [10]. In the studied

case, the equilibrium $\beta'(Al_3Zr)$ phase had provided sufficient and efficient heterogeneous nucleation sites during solidification.

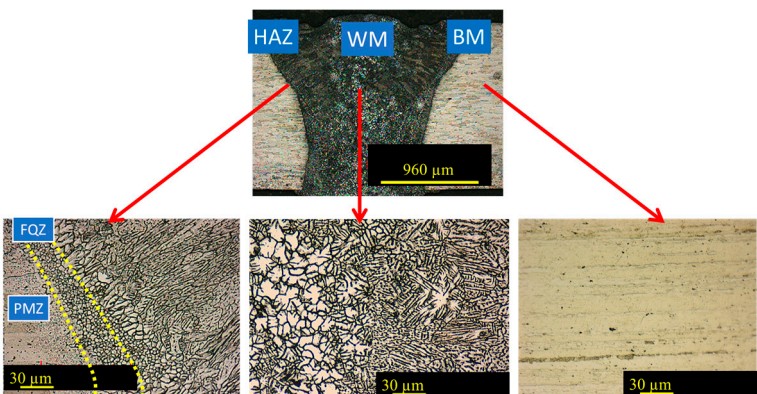

**Figure 2.** Optical microscopy images of the cross-section of the welded joint.

*3.2. Mechanical Properties*

Table 2 presents the key mechanical properties of the base metal and the welded joint, where $\sigma_{UTS}$ is the ultimate tensile strength, $\sigma_{YP}$ is the yield point, $\delta$ is elongation, and $k_i$ is the coefficient showing the ratio of the mentioned parameters for the welded joint to those for the base metal. For each mode, at least three samples were tested. The mean dispersions of the determined values were 2.2% for the ultimate tensile strength, 3.2% for the yield point, and 10% for elongation.

**Table 2.** The key mechanical properties of the base metal and the welded joint.

| Sample | $\sigma_{UTS}$, MPa | $k_1$ | $\sigma_{YP}$, MPa | $k_2$ | $\delta$, % | $k_3$ |
|---|---|---|---|---|---|---|
| Base metal | 556 | – | 514 | – | 10.4 | – |
| Welded joint | 310 | 0.55 | 295 | 0.57 | 0.7 | 0.06 |

Figure 3 shows typical stress–strain diagrams for the base metal, as well as the welded joint both before and after quenching from different temperatures. For the base metal, the Portevin–Le Chatelier (PLC) effect was absent (curve 1), while it was observed after quenching. The PLC effect manifested itself in a sharp localization of plastic strains, causing notches on stress–strain diagrams and changes in the material ductility. This effect was observed upon natural aging of the 2195 Al–Cu–Li alloy [36], which was caused by the joint interaction of dissolved copper and lithium atoms with mobile dislocations. In the studied cases, the character of the PLC effect was changed with the rising quenching temperature: the A type instability of the plastic flow was characteristic for 500 °C, the C type for 560 °C, and a combination of the C and A types for 530 °C.

For the quenched sample, the ultimate tensile strength ($\sigma_{UTS}$), yield point ($\sigma_{YP}$), and elongation ($\delta$) versus quenching temperature dependencies are shown in Figure 4. Quenching increased the ultimate tensile strength from 310 up to 406 MPa (by ~1.29 times). After quenching from 500 °C, the yield point was minimal, but it enhanced with rising quenching temperature, approaching the level of the as-welded joint.

For greater clarity of the dynamics of the mechanical properties depending on the quenching temperature, changes in the ratios of the ultimate tensile strength ($k_1$), the yield point ($k_2$), and elongation ($k_3$) for the welded joint to those for the base metal are shown in Figure 5 for all studied cases. The $k_1$ and $k_2$ coefficients possessed their maximum levels at the quenching temperature of 560 °C, while $k_3$ was the greatest at 500 °C and decreased then. As followed from Figure 4, the $k_1$ and $k_2$ curves were characterized by similar behaviors.

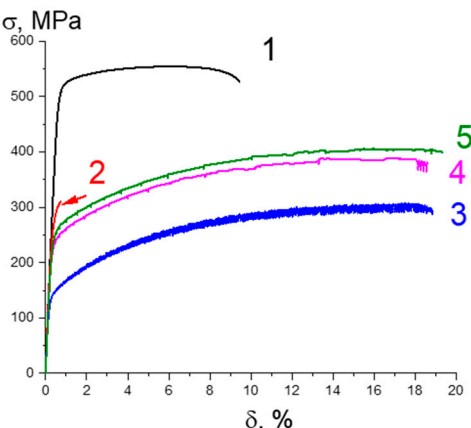

**Figure 3.** The stress–strain diagrams for the base metal (1), as well as for the welded joint before (2) and after quenching from 500 °C (3), 530 °C (4), and 560 °C (5).

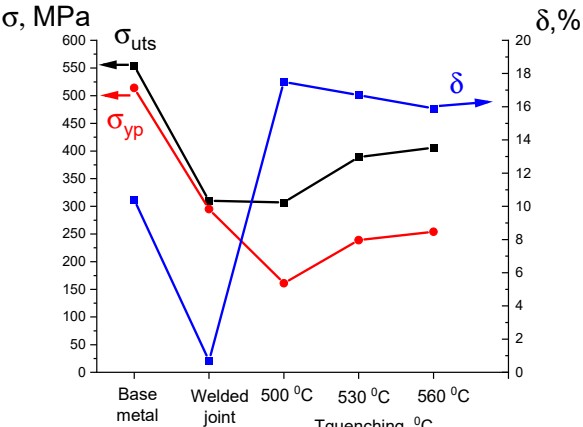

**Figure 4.** The ultimate tensile strength ($\sigma_{UTS}$), yield point ($\sigma_{YP}$), and elongation ($\delta$) versus quenching temperature dependencies for the welded joint.

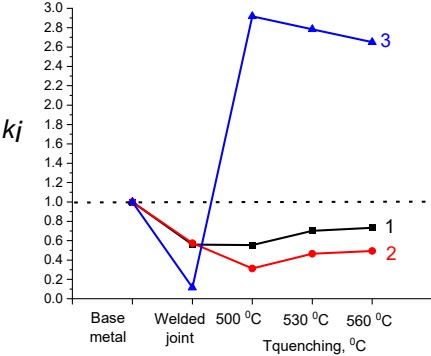

**Figure 5.** The ratios of the ultimate tensile strength (1), yield point (2), and elongation (3) for the welded joint both before and after quenching from different temperatures.

Table 3 shows the artificial aging modes used at a quenching temperature of 530 °C.

Figure 6 shows summarized data on the mechanical properties of the welded specimens (after quenching at the temperature of 530 °C) from the artificial aging temperature and its duration. Diagrams were drawn by approximating the $\sigma_{UTS}$, $\sigma_{YP}$, and $\delta$ mean values from the artificial aging temperature-time characteristics using the least squares method. In order to facilitate understanding, the Z-axis of the ultimate tensile strength, yield strength, and elongation values showed as color maps on the right. The surfaces had areas of maximum (topographic ridge) and minimum (topographic depression).

**Table 3.** Modes of artificial aging at quenching temperature of 530 °C.

| № | Temperature, °C | Time, h |
|---|---|---|
| 1 | 160 | 24 |
| 2 | 160 | 32 |
| 3 | 160 | 40 |
| 4 | 180 | 24 |
| 5 | 180 | 32 |
| 6 | 180 | 40 |

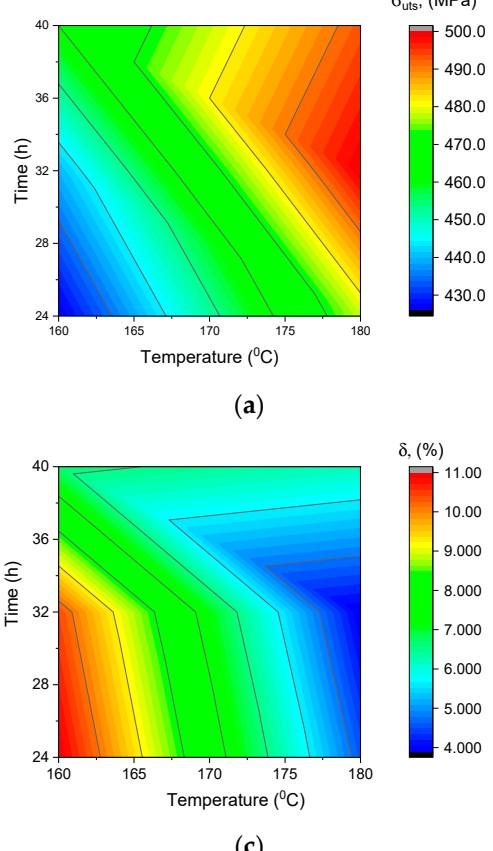

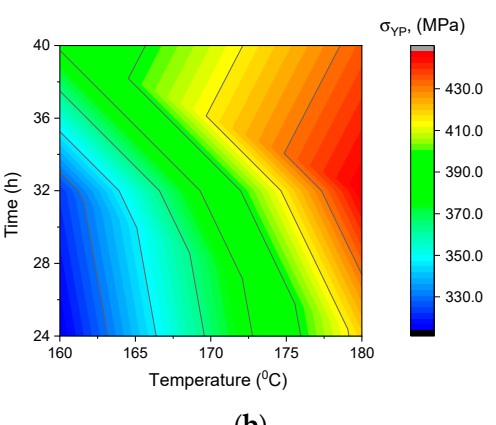

(a)

(b)

(c)

**Figure 6.** 2D diagrams of the mechanical properties of the welded samples vs. the temperature-time parameters of artificial aging: (**a**) $\sigma_{UTS}$, (**b**) $\sigma_{YS}$, (**c**) δ.

In the temperature range from 175–180 °C and in the duration range from 32–36 h, a clear maximum is observed, while the maximum values of the ultimate strength are 500–510 MPa, i.e., 91–92% of the values for the original alloy as delivered. The minimum is reached at an artificial aging temperature of 160 °C and durations of 24–26 h, where the values are 440–450 MPa. The yield point of samples with a welded joint and a pronounced maximum is observed at T = 180 °C and a time of 32–36 h, while the values are ≈450 MPa, i.e., 87% of the values for the original alloy as delivered.

The minimum limit of elongation is reached at T = 160 °C and in the duration range of 24–26 h, while the values are less than 330 MPa. On the 3D surface elongation of specimens with a welded joint, the minimum is reached at T = 180 °C and duration range of 32–36 h, while the values are 4.9–6.2%, i.e., 47–60% of the values for the original alloy as delivered. The maximum relative elongation is reached at T = 160 °C and time 24–26 h, while the values are 10–11%, which is equal to the value for the original alloy in the delivered state. The areas of maximum ultimate tensile strength and yield point, depending on the temperature-time characteristics of artificial aging.

In the areas of maximum ultimate tensile strength and yield point, there is a zone of decrease in ultimate relative elongation (minimum), and vice versa, in areas of minimum tensile strength and yield strength; the relative elongation is maximum.

After PWHT, which included quenching from 530 °C and subsequent artificial aging at 180 °C for 32 h, the welded joint was characterized by the following tensile strength properties: $\sigma_{UTS}$ = 500 MPa, $\sigma_{YP}$ = 450 MPa and $\delta$ = 3.9%. Mechanical properties were low.

The artificial aging regime with maximum mechanical properties was used for a quenching temperature of 560 °C.

Below, the mechanical properties are reported for the welded joint after PWHT, which included quenching from 530 to 560 °C and subsequent artificial aging at 180 °C for 32 h. In particular, stress–strain diagrams are shown in Figure 7, with the same data for the base metal as a reference.

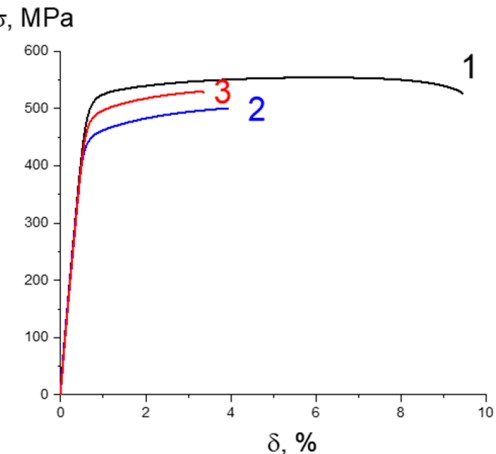

**Figure 7.** The stress–strain diagrams for the base metal (1), as well as for the welded joint after quenching from 530 °C (2) or 560 °C (3) and subsequent artificial aging at 180 °C for 32 h.

However, they were $\sigma_{UTS}$ = 530 MPa, $\sigma_{YP}$ = 485 MPa, and $\delta$ = 3.35% after the quenching temperature rose up to 560 °C. Respectively, the PWHT efficiency depended on the quenching temperature. In the studied cases, its optimal level was 560 °C since the ultimate tensile strength of the welded joint reached 530 MPa, i.e., its ratio to that for the base metal was 0.95. Therefore, the optimal PWHT mode for the studied welded joint was quenched from 560 °C, and subsequent artificial aging was conducted at 180 °C for 32 h.

### 3.3. SEM Examinations

Figures 8 and 9 show SEM images of the microstructures of the base (a, b, c), weld (d, e, f), metals, and HAZ (g, h, i) before and after PWHT at magnifications of 5000× (Figure 8) and 100,000× (Figure 9), respectively. The region shown enlarged in Figure 9 is highlighted in Figure 8 with a red square.

As followed from Figure 8a, the base metal had a typical recrystallized microstructure without any pronounced dendrites. Small (on the order of 20–40 nm) light particles were observed in the solid solution (indicated by red arrows in Figure 9). The microstructure of the weld metal and HAZ was fundamentally different, characterized by many dendrites (Figure 8d). Various SPPs and their agglomerates were located mainly along the grain boundaries (Figure 9d,g). At the grain boundaries, many small light particles 60–150 nm in size were observed. After quenching, the microstructure of the base metal practically did not change, in contrast to the fundamental variations in the weld metal. In this case, no dendrites were found, but a homogeneous solid solution was observed (Figure 8e). After subsequent artificial aging, the microstructures of the base and weld metals became similar (Figure 8c,f). At the nanoscale, light-colored inclusions 40–60 nm in size were observed in the solid solutions (Figure 9c,f,i).

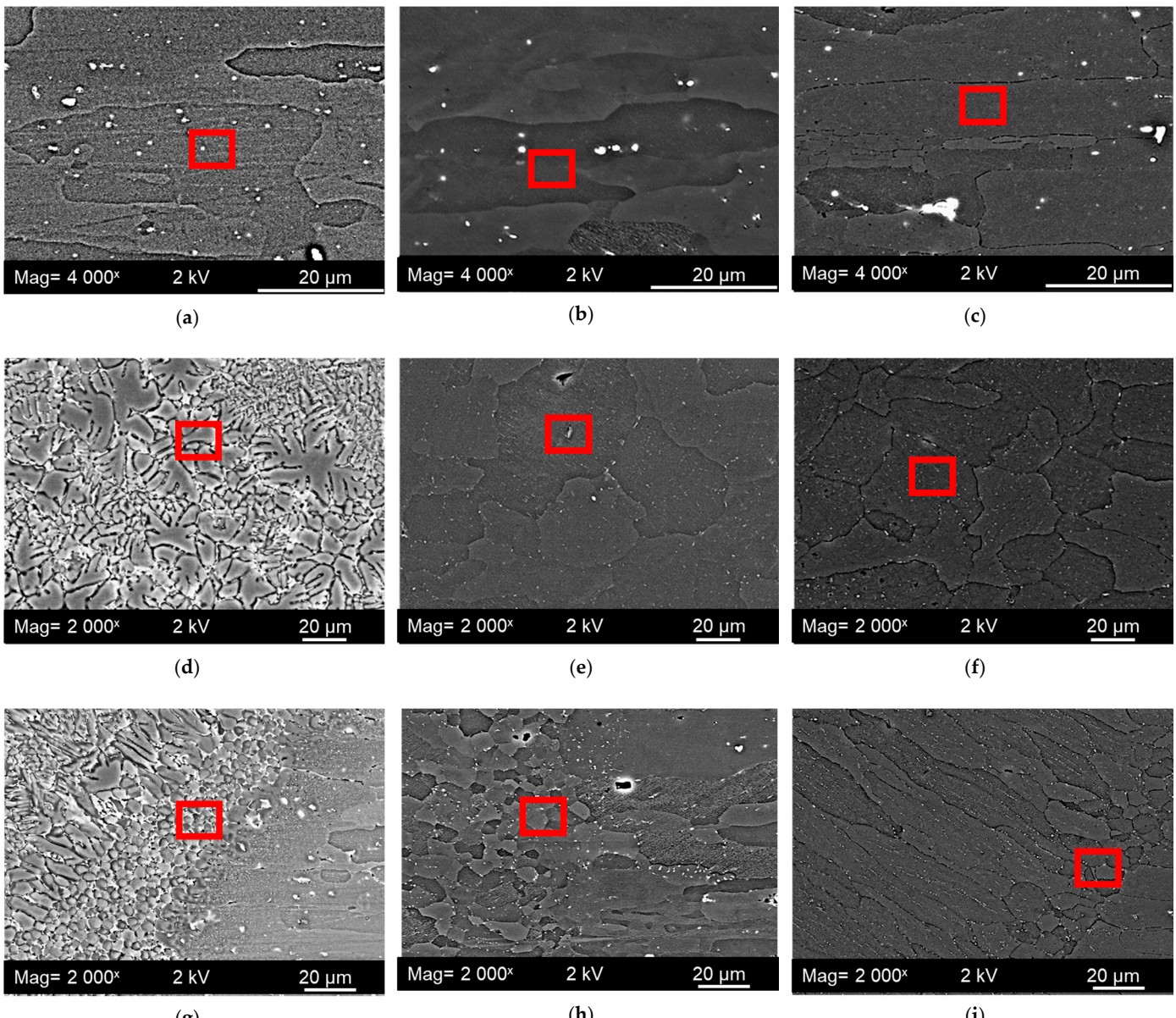

**Figure 8.** The SEM images of the microstructure of the base (**a**–**c**), weld (**d**–**f**), metals, and HAZ (**g**–**i**) before PWHT (**a**,**d**,**g**), after quenching from 560 °C (**b**,**e**,**h**), as well as after quenching from 560 °C and subsequent artificial aging (**c**,**f**,**i**). Magnification of 5000×.

The welded joint fractured through the interface between the weld metal and the HAZ, as reported by other authors [10]. However, PWHT improved all studied mechanical properties. The optimal PWHT resulted in the even microstructure of the weld metal. More detailed study of various zones of the weld metal and HAZ by transmission electron microscopy are ongoing.

EDX analysis of the chemical composition of the base metal showed the presence of the main alloying elements (Figure 10a). At the grain boundaries in the weld metal, there was an increase in the copper concentration up to 6.51 wt.%, but a reduction down to 0.84 wt.% was observed in the solid solution (Figure 10b). After quenching, the concentrations of the alloying elements in the solid solution of the weld metal were restored to the levels of the base metal (Figure 10c).

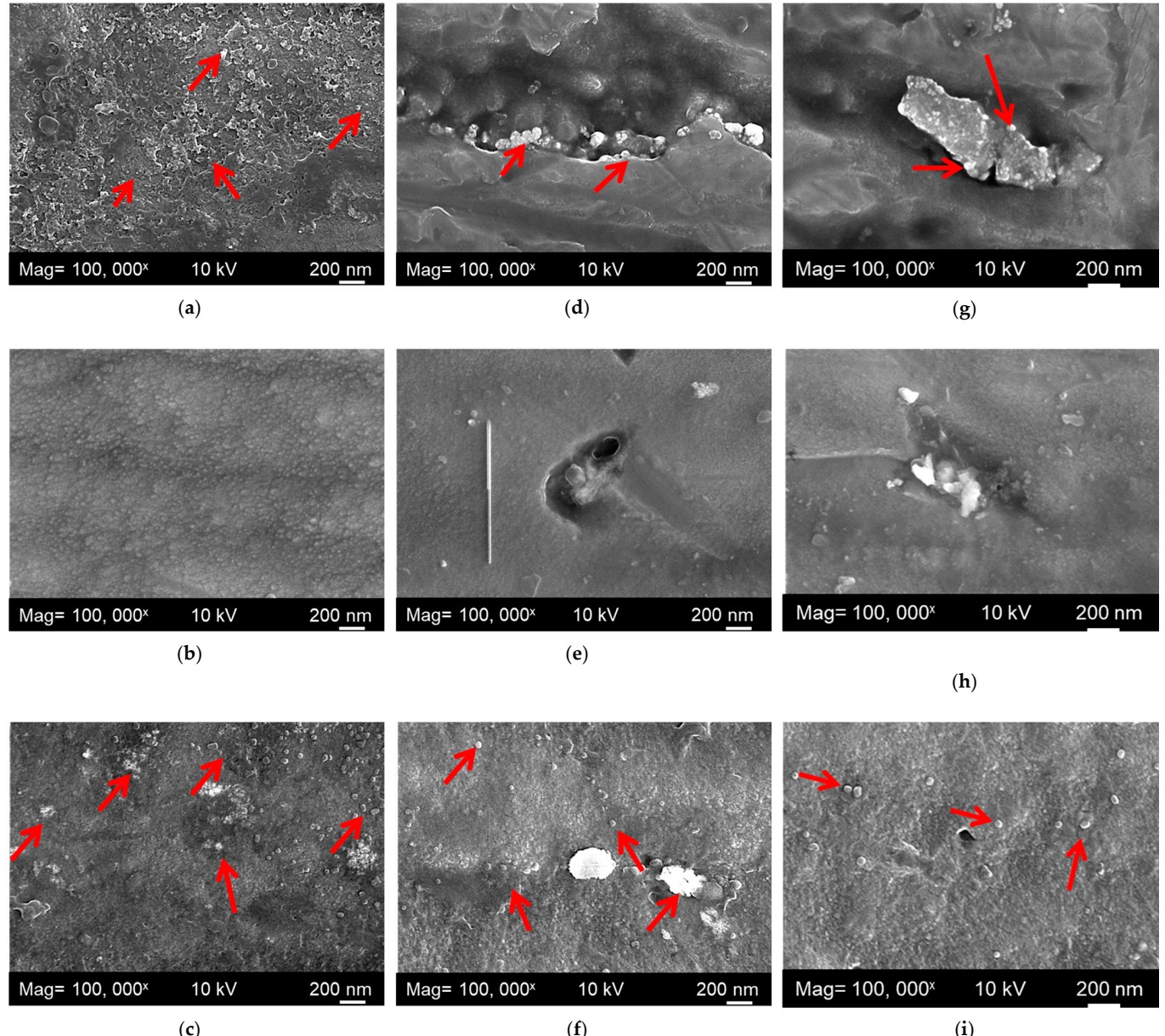

**Figure 9.** The SEM images of the microstructure of the base (**a–c**), weld (**d–f**), metals and HAZ (**g–i**) before PWHT (**a,d,g**) after quenching from 560 °C (**b,e,h**), as well as after quenching from 560 °C and subsequent artificial aging (**c,f,i**). Magnification of 100,000×.

Figure 11 shows X-ray diffractograms of the base metal, HAZ, and weld metal before PWHT, after quenching, and after quenching and subsequent artificial aging. The data were obtained using synchrotron radiation. The small beam area (100 × 400 μm) enabled analyses of the bulk material locally. The base metal included the main α-Al phase with the Fm3m cubic structure. In addition, some additional reflections were observed, in particular the $T_1(Al_2CuLi_3)$ phase at angles of 20.59°, 42.09°, and 48.78°; the $T_2(Al_6CuLi_3)$ phase at 28.77° and 39.96°; the θ′($Al_2Cu$) phase at 20.69°; and the S′($Al_2CuMg$) phase at 40.93° and 55.36°. In the as-welded metal (Figure 11a), the most intense reflections were recorded at angles of about 20°. Quenching from the temperature of 560 °C led to almost complete dissolution of the additional phases and the formation of a solid solution with a super-saturated concentration of the alloying elements. Subsequent artificial aging expanded the spectra of the phase reflections, which connected at angles of 20° and 42° (Figure 11c). It should be noted that the angular width of the reflection in the angle range of 20–21° was

about 0.5° at its half maximum, while the angular distance between the $T_1(Al_2CuLi)$ and $\theta'(Al_2Cu)$ phases was 0.1°.

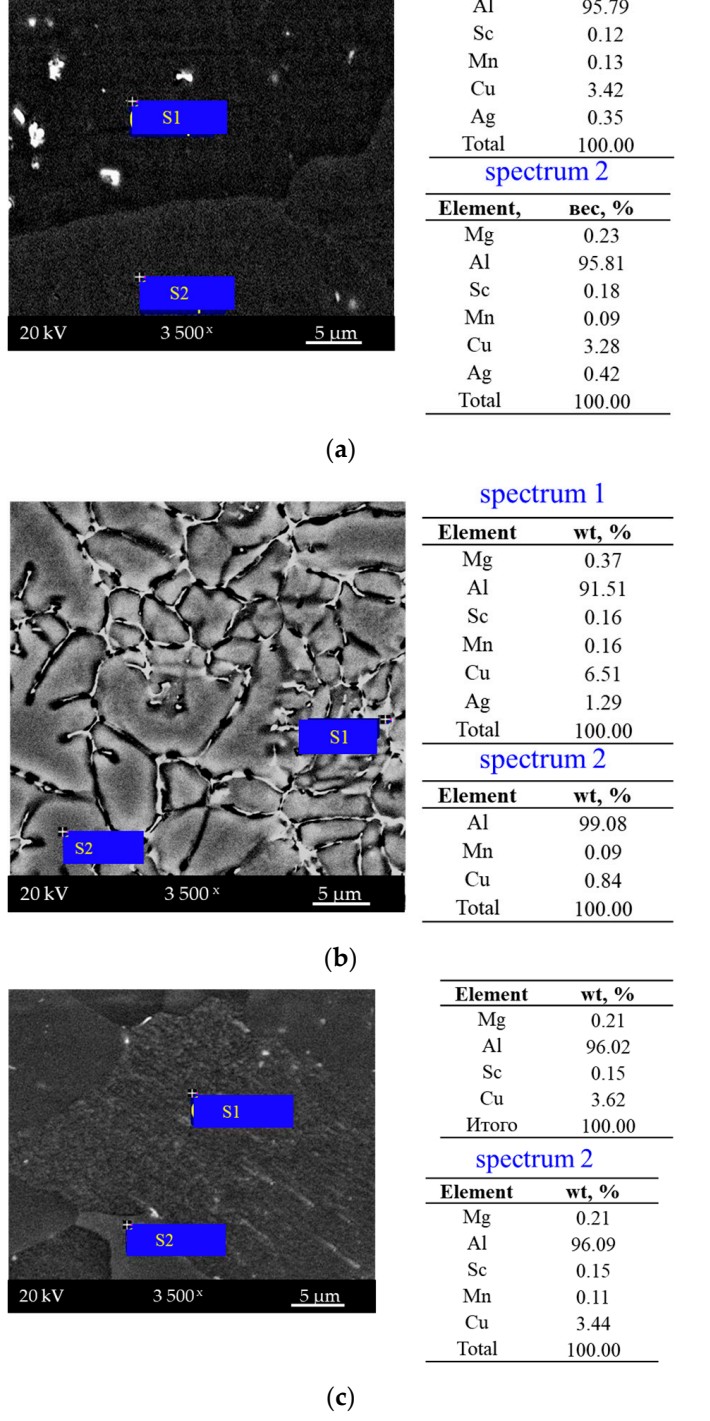

**Figure 10.** SEM images of the microstructure and areas of the chemical composition analysis in the base metal (**a**), as well as in the weld metal before PWHT (**b**) and after quenching (**c**).

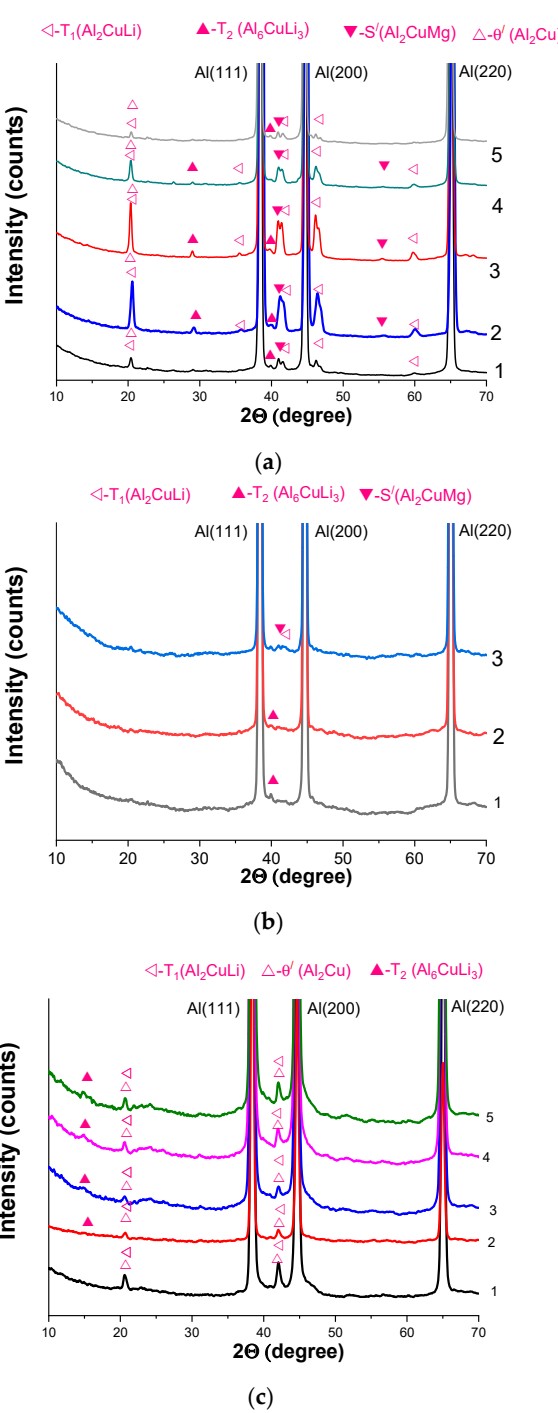

**Figure 11.** The X-ray diffractograms of the base metal, weld metal, and HAZ before PWHT (**a**), after quenching (**b**), as well as after qenching and subsequent artificial aging (**c**). 1, 5—the base metal; 2, 4—the HAZ; 3—the weld metal.

Figure 12 shows reflections of the above phases at the angles of 20–21° and 42–43° with a higher time resolution. According to these data, both base and as-welded metals contained a reflection with a maximum in the region of the $T_1(Al_2CuLi)$ phase. The presence of the $\theta'(Al_2Cu)$ phase could not be ruled out, but its amplitude was negligible. After quenching and subsequent artificial aging, another reflection was observed with a maximum shifted at greater angles to the region of the $\theta'(Al_2Cu)$ phase. The presence of the $T_1(Al_2CuLi)$ phase could not be ruled out, either, but its amplitude became much lower.

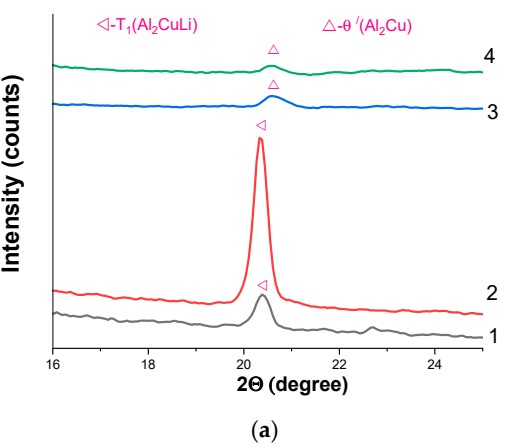
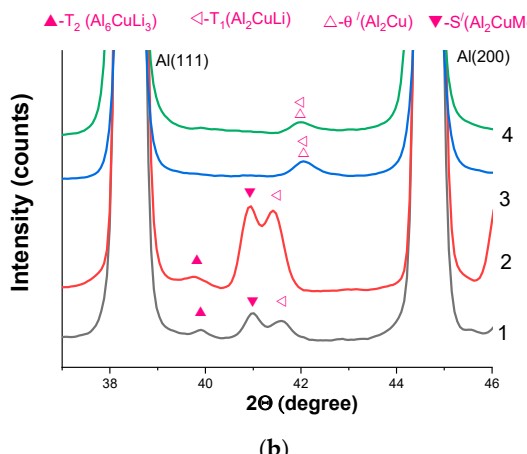

(**a**)　　　　　　　　　　　　　　　　　　　　　(**b**)

**Figure 12.** The X-ray diffractograms of the base metal (1), as well as the weld metal before PWHT (2), after quenching (3), after quenching and subsequent artificial aging (4). The angle ranges of 20–21° (**a**) and 40–42° (**b**).

Figure 13 shows DTA results for the as-welded metal. An increase in the DTA signal was caused by the higher thermal conductivity of the sample compared to that of an empty alundum crucible ($Al_2O_3$) used as a reference. Upon PWHT, the sample was kept at a constant temperature for a certain time. In the DTA process, the temperature changed continuously at a low rate, i.e., both microstructural and phase changes in the solid solution of the weld metal had time to go through all the stages step-by-step, depending on the temperature variations. The as-welded metal curve possessed a complex character, indicating the presence of both exothermic and endothermic processes, the totality of which was expressed as a horizontal straight line in the temperature range from 540–560 °C (Figure 13b). In this interval, homogenization occurred. In the temperature range from 560–580 °C, super-saturated solutions had decomposed (increasing the intensity). Then, the aluminum crystal lattice had ceased to exist due to the main phase of melting.

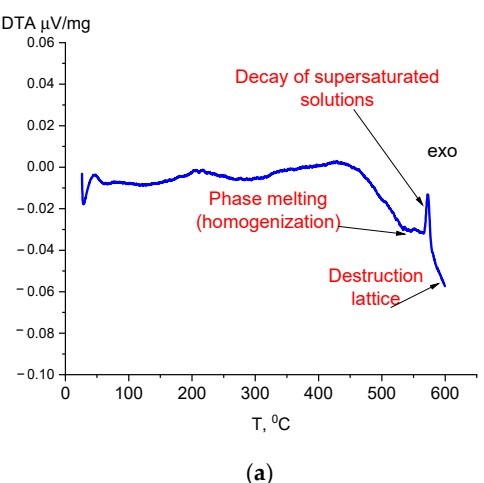
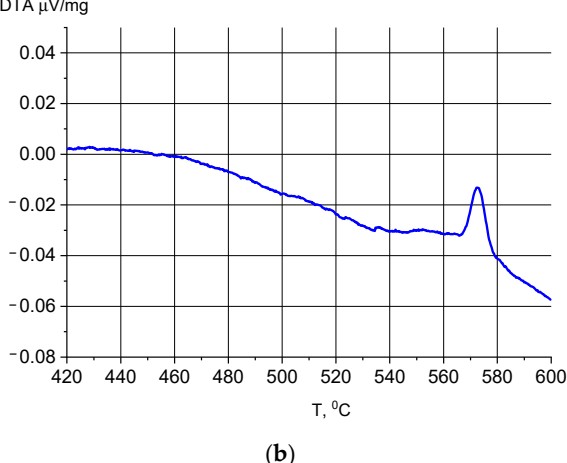

(**a**)　　　　　　　　　　　　　　　　　　　　　(**b**)

**Figure 13.** The DTA curves for the as-welded metal upon its heating. The temperature ranges of 23–600° (**a**) and 420–600° (**b**).

Figure 14 shows changes in the phase composition of the weld metal upon heating of the sample in the furnace. The X-ray patterns were obtained using synchrotron radiation in the in situ mode. The $T_2(Al_6CuLi_3)$ phase (at the angles of 28.77° and 39.96°) dissolved in the temperature range of 450–460 °C. The intensity of the peaks of the $T_1(Al_2CuLi)$, $θ'(Al_2Cu)$, and $S'(Al_2CuMg)$ phases in the temperature range of 540–560 °C decreased,

reflecting their dissolution and the formation of a super-saturated solid solution (SSS). There was a sharp change in the intensity of the main aluminum peaks due to their melting at 560–600 °C. These data on the changes in the phase composition of the weld metal correlated well with the DTA curves shown in Figure 13.

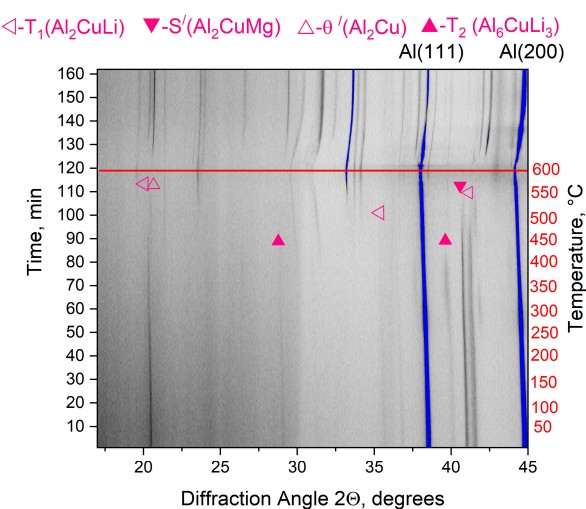

**Figure 14.** The X-ray diffraction patterns for the heated as-welded metal.

Based on the DTA and synchrotron phase analysis results, it could be assumed that the temperature of 560 °C was optimal for the weld metal homogenization, i.e., its quenching. This fact was confirmed by the improvement of the mechanical properties.

## 4. Discussion

During the weld-pool solidification, the thermal cycle significantly affected the microstructure formed at the liquid–solid interface. It was the cooling rate that determined two key parameters, namely the G temperature gradient and the R solidification rate. The cooling rate was related to these parameters: $dT/dt = G \cdot R$, while the G/R ratio determined the morphology of the resulting grain structure. An increase in the cooling rate led to accelerating the nucleation and, respectively, refining the microstructure. The formation of the dendritic structure depended on the G/R ratio. Since laser welding was characterized by a very high R value, the weld metal consisted of dendritic grains.

The results of the phase composition analysis showed that the α-Al phase was the key one in the studied V-1469 alloy, which was the homogeneous solid solution of the alloying elements in aluminum. The use of synchrotron radiation with the spatial resolution of 100 μm enabled the scanning in detail of both the base and weld metals, before and after the optimal PWHT. The obtained data are presented in Figure 15, showing an increase in the reflection of the $T_1(Al_2CuLi)$ phase in the angle range of 20–21° by 2–3 times for the weld metal compared to that for the base metal.

PWHT fundamentally changed the phase compositions in both base and weld metals. Almost complete dissolution of all additional phases and the formation of the solid solution super-saturated with the alloying elements were observed. This fact was confirmed by two independent methods (DTA and XRD in the in situ mode).

The obtained patterns of the phase changes upon welding and PWHT, at first glance, was in dissonance with the variations of the mechanical properties. Figure 16 shows the changes in the $\sigma_{UTS}$ ultimate tensile strength, $\sigma_{YP}$ yield point, and δ elongation values upon laser welding, quenching, and subsequent artificial aging at the optimal PWHT mode. The $\sigma_{UTS}$ of the weld metal was significantly lower than that of the base metal (557 and 310 MPa, respectively). At the same time, the rising content of the hardening $T_1(Al_2CuLi)$ phase was observed. However, no hardening phases were found after quenching from the temperature of 560 °C, but the mechanical properties were maximal. The $\sigma_{UTS}$ value of the

weld metal increased from 310 up to 406 MPa (by ~1.3 times), while the $\sigma_{YP}$ level decreased from 295 down to 254 MPa. Subsequent artificial aging enhanced them more significantly: the $\sigma_{UTS}$ level rose from 406 up to 530 MPa and the $\sigma_{YP}$ value increased from 254 up to 485 MPa. The elongation of the weld metal sharply enhanced up to 15.9% after quenching but lowered down to 3.9% after artificial aging.

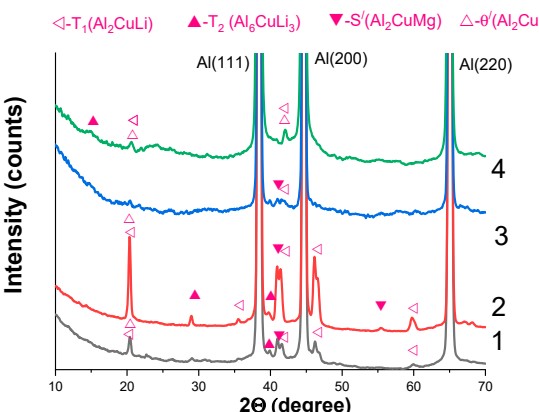

**Figure 15.** The X-ray diffractograms, obtained using synchrotron radiation in the transmission mode, for the base metal (1), as well as for the weld metal before PWHT (2), after qunching (3), after quenching, and subsequent artificial aging (4).

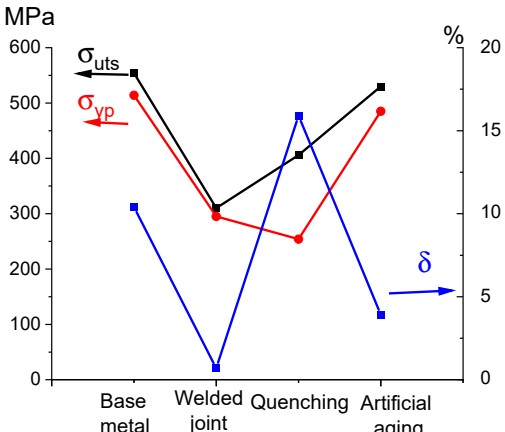

**Figure 16.** The evolution of the mechanical properties of the V-1469 alloy upon laser welding, quenching, and subsequent artificial aging at the optimal PWHT mode.

The welded joint fractured through the interface between the weld metal and the HAZ (FQZ), as reported by other authors [10]. The welding joint Al–Li alloys were very sensitive to fusion line cracking, which is closely related with a FQZ along the fusion line. The narrow FQZ is composed of very fine nondendritic equiaxed grains surrounded by grain boundary eutectic products, due to the poor mechanical properties. The addition of Li can effectively increase the volume fraction of precipitates by decreasing Zr solid solubility in aluminum. Then, the higher temperature phase $Al_3(Li_x,Zr_{1-x})$ is able to form by replacing the atom Zr of $Al_3Zr$ by Li. Compared with the particle $Al_3Zr$, these new ternary particles are expected to provide more potential heterogeneous nucleation sites and have higher nucleation efficiency since the lattice mismatch with the $\alpha$-Al matrix is much lower. Moreover, lithium is a surface-active element that can easily absorb onto the surface of $Al_3Zr$ and $Al_3(Li_x,Zr_{1-x})$. It is beneficial to decrease the interfacial energy between precipitates and growing solids, making great contributions to heterogeneous nucleation. Li is considered the most effective element to reduce the surface tension of liquid aluminum.

Lithium can generally facilitate the formation of FQZ in Al–Li alloy welds with fair finer equiaxed grains based on the classical heterogeneous nucleation theory [10].

In our study, PWHT improved mechanical properties. The width of the HAZ after quenching and artificial aging remained virtually unchanged. However, the number of equiaxed grains decreased (see Figure 8g,h,i).

Various SPPs and their agglomerates were located mainly along the grain boundaries (Figure 9g). After quenching and artificial aging, light-colored inclusions ≈40 nm in size were observed in the solid solutions (Figure 9i). To confirm the composition, density, and distribution of the hardening phases, it is necessary to conduct studies using transmission electron microscopy.

For explaining these conflicting results, a detailed analysis of the processes in the base and weld metals was required. Laser welding fundamentally changed the microstructure of the V-1469 alloy. In particular, all SPPs dissolved, including the $T_1(Al_2CuLi)$ phase. Upon fast cooling of the weld pool, the $\alpha$-Al phase solidified, releasing part of the alloying elements as separate phases at the grain boundaries. The peripheral zones of the dendrite branches were enriched with the elements lowering the melting point of aluminum (in particular, copper). In addition, other phases could be formed, including both non-equilibrium eutectics and intermetallic compounds interacting with aluminum through the eutectic reaction. For example, the $T_1(Al_2CuLi)$ phase had to be located along the grain boundaries and subgrains. In the studied case, the formation of agglomerates of intermetallic particles at the grain boundaries contributed to their contrast in the weld metal (Figure 8a). At the nanoscale, the SEM images showed the significantly lower concentration of fine particles in the solid solution of the weld metal, which confirmed the conclusion about the sharp decrease in the concentration of the $T_1(Al_2CuLi)$ phase inside dendrites. In this way, it was shown for the first time that the spatial distribution of the hardening phases significantly affected the mechanical properties of the welded joint. During the weld-pool solidification, the $T_1(Al_2CuLi)$ phase formed along the grain boundaries. Respectively, its reflection level was higher by 2–3 times in the angle range of 20–21° in the weld metal compared to that in the base metal.

Based on the obtained results, it could be assumed that the precipitation (segregation) of copper led to the formation of the intermetallic $T_1(Al_2CuLi)$ phase at the grain boundaries. This phenomenon was the reason for both low strength and brittle fracture of the welded joint. X-ray diffraction using synchrotron radiation revealed the high concentration of the $T_1(Al_2CuLi)$ phase in the weld metal, reflecting the great reliability of the phase identification.

It should be noted that the lattice parameter was lower in the weld metal (a = 4.0454(5) Å), compared with that in the base metal (a = 4.0519(5) Å) [32]. Typically, reducing the matrix lattice period was associated with rising copper content in solid solutions [37]. It could be assumed that no free copper was preserved in the solid solution of the initial V-1469 alloy after its industrial production route, including thermomechanical processing. Therefore, it was presented only in some intermetallic phases, such as $T_1(Al_2CuLi)$. After laser welding, most copper atoms were concentrated at the grain boundaries. However, about 0.37 at.% Cu, i.e., approximately 1/4–1/5 of its content, remained in the solid solution inside dendrites without the formation of the hardening $T_1(Al_2CuLi)$ phase.

Quenching evened out the mechanical properties of the base and weld metals, i.e., different processes proceeded in them. In the solid solution of the base metal, partial dissolution of the hardening phases occurred, which manifested itself in the nanostructure changes. The eutectic, found at the grain boundaries, was enriched in both copper and magnesium, possessing low melting points [19]. This phenomenon contributed to the acceleration of diffusion processes at the high quenching temperatures (above 500 °C). As a result, quenching from 560 °C led to the weld-metal homogenization with the SSS formation. This fact was also confirmed by the results of both DTA and phase analysis using synchrotron radiation in the in situ mode. In the solid solution, the copper concentration

increased from 0.84 wt.% up to 3.62 wt.%, which was determined by the dissolution of the $T_1(Al_2CuLi)$ phase in the light agglomerates at the grain boundaries.

Previous studies by the authors [38,39], devoted to the phase composition, micro- and nanostructure, as well as functional characteristics of some aluminum–lithium alloys, made it possible to evaluate the role of various mechanisms affecting the mechanical properties of the weld metals. For example, the influence of quenching on the phase composition and the mechanical properties were investigated for the weld metals of the 1420 (Al-5.2Mg-2.1Li) and 1424 (Al-4.9Mg-1.65Li) alloys, i.e., without copper in their compositions. In those cases, both $\sigma_{UTS}$ and $\sigma_{YP}$ values of the welded joints remained almost constant after quenching from 530 °C. These data allowed for the conclusion that the presence of copper in the Al–Cu–Li alloy significantly affected the dependence of the mechanical properties on the quenching temperature.

In general, the purpose of quenching is to obtain extremely non-equilibrium phases in an alloy. Such a state provides a direct increase in both hardness and strength (in comparison with the equilibrium state), as well as the possibility of further hardening upon subsequent aging.

For aluminum alloys, characterized by phase transformations in the solid state (heat treatable grades), the most unstable structure at room temperature is their SSS of alloying components in aluminum, concentrations of which can be higher by 10 times the equilibrium levels. With such a structure, aluminum alloys are also ductile, but much stronger than in the equilibrium state. To achieve the maximum strength of the heat treatable alloys, it is necessary to obtain some intermediate structure through any HT procedures, which corresponds to the initial stages of the SSS decomposition. For example, if the Al–4Cu alloy is heated to a temperature necessary for complete dissolution of the $\theta'(Al_2Cu)$ and $S'(Al_2CuMg)$ phases in aluminum and cooled in water to room temperature, then a solid solution containing 4% copper is retained due to fast cooling, because any intermetallic phases do not have time to form. Since the equilibrium solubility of copper in aluminum at low temperatures is about 0.2%, the solid solution in the quenched Al–4Cu alloy is super-saturated with copper by more than 20 times.

As the temperature rises, the process of the SSS decomposition develops in Al–Cu–Li alloys as follows. When isolating each hardening phase from the SSS, intermediate stages are probable in various combinations, which can be represented in the following way [40]:

For Cu/Li $\geq$ 4.0; $\alpha$ (SSS) $\rightarrow$ Guinier–Preston (GP) zones $\rightarrow$ $\theta''$ $\rightarrow$ $\theta'$
For Cu/Li = 2.5–4.0; $\alpha$ (SSS) $\rightarrow$ GP zones $\rightarrow$ GP zones + $\delta'$ $\rightarrow$ $\theta''$ + $\theta'$ + $\delta'$ $\rightarrow$ $\delta'$ + $T_1$ $\rightarrow$ $T_1$
For Cu/Li = 1.0–2.5; $\alpha$ (SSS) $\rightarrow$ GP zones + $\delta'$ $\rightarrow$ $\theta'$ + $\delta'$ $\rightarrow$ $\delta'$ + $T_1$ $\rightarrow$ $T_1$
For Cu/Li $\leq$ 1.0; $\alpha$ (SSS) $\rightarrow$ GP zones $\rightarrow$ $\delta'$ + $T_1$ $\rightarrow$ $T_1$

It follows from the presented patterns that the probability of the $\theta'$ phase formation increases with the rising Cu/Li ratio.

This study enabled the justification of patterns of the $\theta'$ phase formation upon laser welding and PWHT, i.e., under nonequilibrium conditions. The fact of the formation of SPPs at the grain boundaries after melting, and their dissolution at high quenching temperatures, was already known. However, the dissolution of the $T_1(Al_2CuLi)$ phase in the weld metal upon quenching from the temperature of 560 °C was observed for the first time. This fact was confirmed by two independent methods (DTA and synchrotron radiation in the in situ mode). As a result of artificial aging, the $\theta'(Al_2Cu)$ phase was formed.

It should be noted that the order changes, as a very local process of the formation of both GP zones and $\theta''$ phases, depend only on a few jumps per atom, while the $T_1(Al_2CuLi)$ + $\theta'(Al_2Cu)$ conglomerates are determined by long-range atomic diffusion processes. It is important that the formation of the GP zones is associated with submicroscopic regions with an increased copper content in the solid solution. For example, if a solid solution contains 4% copper, and the $\theta'(Al_2Cu)$ compound, which may be separated from the solid solution, contains 52% copper, then its content is intermediate in the GP zones, increasing as the process develops. In Al–Cu alloys, the GP zones possess a lamellar shape, which have been formed on the (100) crystallographic planes and are part of the solid solution.

Their crystal structure is the same as that of the solid solution, but the lattice constant is smaller due to the increased copper content, the atomic radius of which is less than that for aluminum. The GP zones are also characterized by small dimensions (thicknesses of 0.5–1.0 nm and diameters of 4–10 nm). At the next stage, precipitates are formed in the solid solution (for example, the intermediate $\theta''$ one, the composition of which corresponds to the $\theta'(Al_2Cu)$ phase). It is assumed that the $\theta''$ phase nucleates and grows in the GP zones due to the presence of a stress field around them. In [40–45], the behavior of precipitates around dislocations has been reported. In particular, the metastable phases have a strong tendency to precipitate within a small volume of the dislocation core where the aluminum matrix is under compressive stress.

The $\theta''$ phase is fully coherent with the aluminum solid solution [40,46,47]. It is characterized by an ordered mutual arrangement of copper and aluminum atoms, in which certain parts of the planes are occupied by only one of them. The maximum thicknesses of the $\theta''$ phase precipitates are 10 nm, but their diameters are up to 150 nm. It should be noted that such particles may be called a phase only conditionally since they do not have a discrete interface with the matrix, which significantly complicates their detection by X-ray fluorescence technique.

As a result of quenching, an improvement of the mechanical properties is apparently associated with the beginning of the SSS decomposition with the formation of the GP zones and the precipitation of the intermediate metastable $\theta''$ phase (in Al–Cu alloys).

It is known [46,47] that the high ultimate tensile strength may be achieved by the well-balanced formation of both coherent and subsequent semi-coherent precipitates. After quenching, significant hardening of an alloy is caused by the fact that the GP zones and metastable particles of intermetallic phases serve as an obstacle to the movement of dislocations. During plastic deformation, moving dislocations cut both GP zones and metastable particles of the $\theta''$ phase. Nevertheless, elastic strains around the GP zones and disruption of the order in the arrangement of atoms (when dislocations passed through the GP zones) increases the stress level required for the movement of dislocations.

For the Al–Cu–Mg alloy, diffusion processes in a solid solution and their influence on the formation of stable phases have been reported in detail [47]. Therefore, the difference between the two processes should be emphasized. Upon the cooling of the weld pool, the formation of the $T_1(Al_2CuLi)$ and $S(Al_2CuMg)$ phases at the boundaries of subgrains may be caused by nonequilibrium solidification, i.e., diffusion processes in the liquid phase. Upon quenching and subsequent artificial aging, phase transformations occur in the solid solution, i.e., significantly slower.

The temperature dependence of the $D$ diffusion coefficient is described as $D = D_0 \cdot \exp(-Q/RT)$, where $D_0$ is a constant value characterizing a system; $Q$ is the activation energy, which determines its amount required to pull an atom out of a crystal lattice; $R$ is the universal gas constant; and $T$ is the absolute temperature. Respectively, the diffusion coefficient decreases with increasing the activation energy, but it enhances with rising temperatures. Considering the artificial aging duration at a temperature of 170 °C, the characteristic time for the formation of hardening phases is 48 h [47].

Precipitation of the intermediate $\theta'$ phase particles is actually the beginning of the SSS decomposition. The $\theta'$ phase composition corresponds to that for stable $\theta(Al_2Cu)$, but it possesses another crystal lattice. It is intermediate in the sense that it matches the aluminum lattice more easily than that of the $\theta$ phase. The $\theta'$ phase precipitates are formed from the $\theta''$ ones, and they are conjugated and coherent with the aluminum lattice along the (100) planes. Thus, the $\theta'$ phase is not completely separated from the matrix by the interface. When the stable $\theta(Al_2Cu)$ phase is formed, the lattice coherence of the matrix and the precipitating phase are completely violated.

Intermetallic particles with a crystal lattice different from that of the matrix and an ordered arrangement of atoms represent a more serious obstacle to the movement of dislocations. In this case, dislocations do not cut but bypass these particles, forming loops and accumulating dislocations [46–48]. As the distance between particles decreases upon

aging, the stress level required to bend the dislocations and push them between the particles increases, raising the yield point. That is why the maximum strengthening effect is observed under those aging conditions in which dispersed intermetallic particles are formed that are uniformly distributed in the matrix at small distances from one another. Coarseness of the particles and an increase in the distance between them reduces the strength characteristics. Typically, the goal of an optimization of the artificial aging parameters (both temperature and duration) is to improve the mechanical properties. The presented physical patterns of the weld metal hardening fully correlate with the above experimental results and are in good agreement with the previously reported data on the effect of quenching and artificial aging of the V-1469 alloy [47,49]. According to [50,51], many phases remain in a matrix at the grain boundaries if the quenching temperature is low. It reduces the supersaturation degree in a solid solution and weakens the quenching effect. On the contrary, more phases dissolve with rising the quenching temperature, improving the mechanical properties. In this case, the role of GP zones can be understood from the copper-depending results of the quenching of aluminum alloys. For example, the absence of copper in Al–Mg–Li alloys, i.e., without the formation of the GP zones, quenching does not affect the mechanical properties, and its role is narrowed only to the homogenization of solid solutions and the dissolution of alloying elements [50]. In this case, hardening is achieved by subsequent artificial aging only.

In this study, the maximum mechanical properties of the welded joint have been achieved after the optimal PWHT procedure that included both quenching and subsequent artificial aging. The phase changes upon laser welding, as well as PWHT, are summarized in Table 4. The main phases are those that directly affect the mechanical properties.

**Table 4.** The phase compositions of the base and weld metals.

| | | Initial State | Quenching from 560 °C | Quenching from 560 °C and Subsequent Artificial Aging at 180 °C for 32 h |
|---|---|---|---|---|
| Base metal | Main phases | $\alpha$-Al + $T_1$(Al$_2$CuLi) | $\alpha$-Al + SSS + GP zones + $\theta''$ | $\alpha$-Al + $\theta'$(Al$_2$Cu) |
| | Additional phases | $T_2$(Al$_6$CuLi$_3$) S(Al$_2$CuMg) $\theta'$(Al$_2$Cu) | | $T_1$(Al$_2$CuLi) |
| Weld metal | Main phases | $\alpha$-Al + $T_1$(Al$_2$CuLi)) | $\alpha$-Al + SSS + GP zones + $\theta''$ | $\alpha$-Al + $\theta'$(Al$_2$Cu) |
| | Additional phases | S(Al$_2$CuMg) $T_2$(Al$_6$CuLi$_3$) $\theta'$(Al$_2$Cu) | | $T_1$(Al$_2$CuLi) |

The evolution of the phase composition in the weld metal of the V-1469 alloy can be represented as follows:

$$\text{welding} \rightarrow \qquad \text{quenching} \rightarrow \qquad \text{artificial aging}$$
$$\alpha\text{-Al} + T_1 + \theta' \qquad \alpha\,(\text{SSS}) \rightarrow \text{GP zones} \rightarrow \theta'' \qquad \alpha\text{-Al} + \theta' + T_1$$

Based on the above, it can be stated that the changes in the mechanical properties after quenching from the optimal temperature of 560 °C is caused by the transformation of the microstructure and nanostructure. In particular, the dissolution of agglomerates at the grain boundaries and homogenization of the solid solution causes the beginning of the SSS decomposition with the formation of GP zones and the release of the intermediate metastable $\theta''$ phase (in the Al–Cu alloys). After the optimal artificial aging, new both microstructure and phase composition is observed in the weld metal, which includes mainly the hardening $\theta'$ phase. The presented results provide a comprehensive picture of the changes in the mechanical properties of the joint of the V-1469 alloy, occurring upon laser welding and PWHT (quenching and subsequent artificial aging). The solid solution

parameters of the weld metal (microstructure and nanostructure, the distribution of the alloying elements, the formation and dissolution of the main hardening phases) determines the variations of the mechanical properties.

## 5. Conclusions

In the study, the PWHT mode was optimized for the butt laser-welded joint of the Al–Cu–Li alloy, improving its mechanical properties. Before and after PWHT, both microstructures and phase compositions have been examined by OM and SEM techniques, as well as synchrotron X-ray diffractometry. Based on the obtained results, the following conclusions could be drawn:

- In laser welding, copper atoms were concentrated at the grain boundaries, causing the formations of the $T_1(Al_2CuLi)$, $\theta'(Al_2Cu)$, and $S'(Al_2CuMg)$ phases. As a result, the ultimate tensile strength and yield point of the welded joint were 55% and 54%, respectively, of the levels inherent in the base metal.
- In situ X-ray diffraction analysis on a synchrotron radiation source revealed the dissolution of the $T1(Al_2CuLi)$ and $\theta'(Al_2Cu)$ phases at temperatures of 550–560 °C.
- Quenching from the temperature of 560 °C evenly distributed the alloying elements and formed the SSS, GP zones, and the metastable $\theta''$ phase. As a result, the ultimate tensile strength increased from 306 up to 403 MPa (by ~1.3 times), while the yield point enhanced from 150 up to 250 MPa (by 1.66 times).
- Artificial aging at 190 °C for 32 h resulted in the formation of the main $\theta'(Al_2Cu)$ phase with the additional $T_1(Al_2CuLi)$, one in the weld metal. In this case, the ultimate tensile strength increased from 403 up to 472 MPa (by ~1.17 times), while the yield point enhanced from 259 up to 367 MPa (by 1.46 times).
- The optimal PWHT enabled for the improve of the ultimate tensile strength, yield point, and elongation of the welded joint up to 530 MPa, 485 MPa, and 3.9%, respectively, i.e., 95%, 94%, and 34% from the levels of the base metal.

**Author Contributions:** Conceptualization, A.M.; methodology, A.M., E.K., K.K. and A.S.; investigation, A.M., E.K., K.K. and A.S.; curation, A.M.; writing—original draft preparation, A.M.; supervision A.M.; project administration, A.M.; funding A.M. All authors have read and agreed to the published version of the manuscript.

**Funding:** This research was funded by Russian Scientific Foundation, grant number 23-79-00037 (https://rscf.ru/project/23-79-00037/, accessed on 31 July 2023).

**Data Availability Statement:** Not applicable.

**Acknowledgments:** Part of the work was done at the shared research center SSTRC on the basis of the VEPP-4—VEPP-2000 complex at BINP SB RAS.

**Conflicts of Interest:** The authors declare no conflict of interest.

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
