# Peer review of "Influence of Quenching and Subsequent Artificial Aging on Tensile Strength of Laser-Welded Joints of Al–Cu–Li Alloy"

_metals, doi:10.3390/met13081393_

Round 1

Reviewer 1 Report

This paper reports the effect of PWHT on transverse tensile properties of laser welded Al-Cu-Li alloy, in relation to the dissolution-precipitation sequence during PWHT. Although it may provide some practically useful information to limited extent, but the results and discussions are not much different from classical textbook knowledge on precipitation-hardenable Al alloys. Furthermore, the manuscript is not well written/organized missing necessary experimental information. Because transverse tensile properties of joint are governed by the weakest/softest region, microstructural change in each of FZ, HAZ and base metal region of the welded joint during the PWHT should be explained more exactly and discussed the microstructure of the weakest/softest location in the joint with its transvers tensile properties, so as to highlight the research characteristics of laser welded joint.

Comments;

1.        Generally a laser welded joint consists of FZ, HAZ and base metal region as mentioned on lines 159-160, and the term “weld metal” corresponds to “FZ”, but “weld metal” in Fig. 10 seems to be misused as if “weld metal” included the HAZ and base metal. So as not to confuse the readers, terms “weld”, “welded joint”, “weld metal”, “as-welded metal” and “joint” should be exactly defined and carefully distinguished through the manuscript. Additionally, terms “grain boundary” and “dendrite boundary” should be exactly distinguished through the manuscript.   

2.        A laser welded joint contains heterogeneous microstructure distribution in FZ, HAZ and base metal region. Tensile properties of the joint are usually governed by the weakest/softest region which preferentially deforms and fractures during transverse tensile testing of the joint. But the preferentially deformed region and fracture location in the joint during the transverse tensile test are not clearly shown in this manuscript. Although the microstructure seems to be shown mainly in FZ by SEM in this manuscript, the microstructural change in each of FZ, HAZ and base metal region of the welded joint during the PWHT should be shown more exactly and discussed the microstructure of the weakest/softest location in the joint with its transvers tensile properties, so as to highlight the research characteristics of laser welded joint.

3.        The initial state of the base metal is not described. The heat treatment condition which the as-received base metal had experienced before welding must be exactly indicated.

4.        As far as known from “The applied PWHT procedures included quenching in water after exposure in a muffle furnace at temperatures of 500, 530 and 560 °C for 30 minutes, as well as subsequent artificial aging at a temperature of 180 °C for 32 hours.” on lines 117-118, the PWHT parameter variation is not wide, especially only one parameter set (180 °C for 32 h) for the artificial aging. It is necessary to explain exactly why such a small number of PWHT parameter sets with limited variation is sufficient for optimizing the maximum tensile properties of the present joint.

5.        The reason for the quenching temperature dependent “Portovin-le-Chatelier effect” in Fig. 3 needs to be stated more clearly.

6.        “red arrows” on line 224 cannot be seen in “Figure 7, a” on line 222.

7.        Figure 6 shows that the elongation (δ) for the joint quenched from 530 ºC (2) is larger than that for the joint quenched from 560 ºC (3), while the same “δ=3.9%” for both joints (2) and (3) is written on “lines 210-213 in the text. This discrepancy should be explained.

8.        The schematic illustration in Figure 16 is not well supported by experimental data, because size, shape, density and distribution of these precipitates may not be known from the present experiments.   

9.        There are too many places where numbers that should be subscripted are not subscripted, for examples, T1(Al2CuLi) on lines 25 & 46, Al2Cu on lines 47 & 70, Mg2Si on line 56, Al11Cu5Mn3 on line 61, Al7Cu2Fe on line 61, T1(Al2CuLi) on lines 63 & 322, Al7Cu4Li on line 63, R(Al5CuLi3) on line 63, T2(Al6CuLi3) on line 67, CO2 on line 98, Al3Zr on line 170, k3 on line 194, T1(Al2CuLi3) on line 256, etc.

10.     “C”s should be revised to “ºC”s on lines 205 & 220.

11.     It should be mentioned why three photos (a, b, c) in each of Figs. 1 and 2 are shown for the base metal. It is necessary to mention which region/zone Figs. 1d, e ,f and Figs. 2d, e, f are taken from within the joint. “the weld metal (Figure 7, a)” is a bit strange to me, because Fig. 7a is for the base metal.

12.     “aging” and “ageing” coexist in the text.

Author Response

This paper reports the effect of PWHT on transverse tensile properties of laser welded Al-Cu-Li alloy, in relation to the dissolution-precipitation sequence during PWHT. Although it may provide some practically useful information to limited extent, but the results and discussions are not much different from classical textbook knowledge on precipitation-hardenable Al alloys. Furthermore, the manuscript is not well written/organized missing necessary experimental information. Because transverse tensile properties of joint are governed by the weakest/softest region, microstructural change in each of FZ, HAZ and base metal region of the welded joint during the PWHT should be explained more exactly and discussed the microstructure of the weakest/softest location in the joint with its transvers tensile properties, so as to highlight the research characteristics of laser welded joint.

Response:

  • The authors are very grateful to the Reviewer for the spent time and participation. All changes are highlighted in blue.
  • This study is devoted to changes in the microstructure and phase composition of the weld metal of the Al-3.9Cu-0.3Mg-1.2Li alloy through its post-weld heat treatment (PWHT). The PWHT procedure has included quenching and subsequent artificial aging, affecting the mechanical properties of the joint. The principal feature of the study is that PWHT has been carried out for welded samples, which are significantly inhomogeneous in terms of their microstructure and phase compositions, as well as the mechanical properties. In such cases, conventional HT procedures, used for manufacturing rolled Al-Cu-Li alloys, are not applicable. So, it has been necessary to optimize PWHT modes, since the phase composition of the base metal should not be changed but hardening phases should be formed in the weld metal. The welded joint fractured through the interface between the weld metal and the HAZ, as reported by other authors [doi:10.1016/j.engfracmech.2019.01.013]. So, the authors have not shown photos of the fractured samples. However, PWHT improved all studied mechanical properties.

Comments:

  1. Generally a laser welded joint consists of FZ, HAZ and base metal region as mentioned on lines 159-160, and the term “weld metal” corresponds to “FZ”, but “weld metal” in Fig. 10 seems to be misused as if “weld metal” included the HAZ and base metal. So as not to confuse the readers, terms “weld”, “welded joint”, “weld metal”, “as-welded metal” and “joint” should be exactly defined and carefully distinguished through the manuscript. Additionally, terms “grain boundary” and “dendrite boundary” should be exactly distinguished through the manuscript.  
  • Response. The authors thank the reviewer for the comment. All terms in the text and figures have been consistent with AWS A3.0-2001 Standard welding terms and definitions. The term ‘grain boundary’ has been unified as well.
  1. A laser welded joint contains heterogeneous microstructure distribution in FZ, HAZ and base metal region. Tensile properties of the joint are usually governed by the weakest/softest region which preferentially deforms and fractures during transverse tensile testing of the joint. But the preferentially deformed region and fracture location in the joint during the transverse tensile test are not clearly shown in this manuscript. Although the microstructure seems to be shown mainly in FZ by SEM in this manuscript, the microstructural change in each of FZ, HAZ and base metal region of the welded joint during the PWHT should be shown more exactly and discussed the microstructure of the weakest/softest location in the joint with its transvers tensile properties, so as to highlight the research characteristics of laser welded joint.
  • Response.  It has been added to the manuscript: ‘The welded joint fractured through the interface between the weld metal and the HAZ, as reported by other authors [doi:10.1016/j.engfracmech.2019.01.013]. However, PWHT improved all studied mechanical properties.’ The optimal PWHT resulted in the even microstructure and microhardness of the weld metal. More detailed study of various zones of the weld metal and HAZ by transmission electron microscopy are ongoing. Their results will be presented in a next paper.
  1. The initial state of the base metal is not described. The heat treatment condition which the as-received base metal had experienced before welding must be exactly indicated.
  • Response. The alloy is produced by All-Russian Scientific Research Institute of Aviation Materials [doi:10.1007/s11041-007-0049-y, doi:10.1134/S0036029513090139]. In the production routes, the heat treatment conditions are trade secrets of this company.
  1. As far as known from “The applied PWHT procedures included quenching in water after exposure in a muffle furnace at temperatures of 500, 530 and 560 °C for 30 minutes, as well as subsequent artificial aging at a temperature of 180 °C for 32 hours.” on lines 117-118, the PWHT parameter variation is not wide, especially only one parameter set (180 °C for 32 h) for the artificial aging. It is necessary to explain exactly why such a small number of PWHT parameter sets with limited variation is sufficient for optimizing the maximum tensile properties of the present joint.
  • Response. The authors carried out many trial attempts for optimizing the artificial aging parameters in wide ranges of temperatures and durations. In order not to overload the paper, this data was not included.
  1. The reason for the quenching temperature dependent “Portovin-le-Chatelier effect” in Fig. 3 needs to be stated more clearly.
  • Response. It has been added to the manuscript: ‘The PLC effect manifested itself in a sharp localization of plastic strains, causing notches on stress–strain diagrams, and changes in the material ductility. This effect was observed upon natural aging of the 2195 Al-Cu-Li alloy [doi: 10.1016/j.matchar.2021.111694], which was caused by the joint interaction of dissolved both copper and lithium atoms with mobile dislocations.
  1. “red arrows” on line 224 cannot be seen in “Figure 7, a” on line 222.
  • Response. It has been changed in the manuscript: ‘…indicated by red arrows in Figure 8.
  1. Figure 6 shows that the elongation (δ) for the joint quenched from 530 ºC (2) is larger than that for the joint quenched from 560 ºC (3), while the same “δ=3.9%” for both joints (2) and (3) is written on “lines 210-213 in the text. This discrepancy should be explained.
  • Response. Thanks a lot for the comment; it was a mistake. It has been changed in the manuscript: ‘After PWHT, which included quenching from 530 °C and subsequent artificial aging, the welded joint was characterized by the following tensile strength properties: σUTS=500 MPa, σYP=450 MPa and δ=3.9%. However, they were σUTS=530 MPa, σYP=485 MPa and δ=3.35% after rising the quenching temperature up to 560 °C
  1. The schematic illustration in Figure 16 is not well supported by experimental data, because size, shape, density and distribution of these precipitates may not be known from the present experiments.   
  • Response.The authors agree that the schematic illustration is not fully supported by the experimental data. To confirm the size, shape, density and distribution of the hardening phases, it is necessary to conduct studies using transmission electron microscopy. This study is are ongoing and its results will be presented in a next paper.
  1. There are too many places where numbers that should be subscripted are not subscripted, for examples, T1(Al2CuLi) on lines 25 & 46, Al2Cu on lines 47 & 70, Mg2Si on line 56, Al11Cu5Mn3 on line 61, Al7Cu2Fe on line 61, T1(Al2CuLi) on lines 63 & 322, Al7Cu4Li on line 63, R(Al5CuLi3) on line 63, T2(Al6CuLi3) on line 67, CO2 on line 98, Al3Zr on line 170, k3 on line 194, T1(Al2CuLi3) on line 256, etc.
  • Response. Agree with remark; such errors have been corrected in the text.
  1. “C”s should be revised to “ºC”s on lines 205 & 220.
  • Response. Agree with remark; such errors have been corrected in the text.
  1. It should be mentioned why three photos (a, b, c) in each of Figs. 1 and 2 are shown for the base metal. It is necessary to mention which region/zone Figs. 1d, e ,f and Figs. 2d, e, f are taken from within the joint. “the weld metal (Figure 7, a)” is a bit strange to me, because Fig. 7a is for the base metal.
  • Response. Agree with remark; such errors have been corrected in the text.
  1. “aging” and “ageing” coexist in the text.
  • Response. Agree with remark; such errors have been corrected in the text.

Reviewer 2 Report

This paper investigated the laser welding of Al-Cu-Li alloys with quenching and subsequent artificial aging. This paper is well-structured and interesting. Some comments need to be addressed before publication in Metals, as follows:

(1)    For Al-Cu-Li alloys, another effective welding method is friction stir welding with high quality joints. In introduction, the authors should add some discussion about FSW with referenced to https://doi.org/10.1016/j.pmatsci.2020.100706, https://doi.org/10.1007/s40195-022-01444-0

(2)    The primary difficulties of Al-Cu-Li alloys should be stated and discussed.

(3)    Provide the surface formation of laser welded joints.

(4)    Please rewritten the conclusions.

(5)    The language should be carefully polished since there are several grammar errors and typos.

This paper investigated the laser welding of Al-Cu-Li alloys with quenching and subsequent artificial aging. This paper is well-structured and interesting. Some comments need to be addressed before publication in Metals, as follows:

(1)    For Al-Cu-Li alloys, another effective welding method is friction stir welding with high quality joints. In introduction, the authors should add some discussion about FSW with referenced to https://doi.org/10.1016/j.pmatsci.2020.100706, https://doi.org/10.1007/s40195-022-01444-0

(2)    The primary difficulties of Al-Cu-Li alloys should be stated and discussed.

(3)    Provide the surface formation of laser welded joints.

(4)    Please rewritten the conclusions.

(5)    The language should be carefully polished since there are several grammar errors and typos.

Author Response

This paper investigated the laser welding of Al-Cu-Li alloys with quenching and subsequent artificial aging. This paper is well-structured and interesting. Some comments need to be addressed before publication in Metals, as follows:

  • Response. The authors are very grateful to the Reviewer for the spent time and participation. All changes are highlighted in blue.

(1)    For Al-Cu-Li alloys, another effective welding method is friction stir welding with high quality joints. In introduction, the authors should add some discussion about FSW with referenced to https://doi.org/10.1016/j.pmatsci.2020.100706, https://doi.org/10.1007/s40195-022-01444-0

  • Response. It has been added to the manuscript: ‘It should also be noted another effective method of joining Al-Cu-Li alloys, namely friction stir welding [https://doi.org/10.1016/j.pmatsci.2020.100706, https://doi.org/10.1007/s40195-022-01444-0].

(2)     The primary difficulties of Al-Cu-Li alloys should be stated and discussed.

  • Response. It has been added to the manuscript: ‘After laser welding, the ultimate tensile strength of the joints is about 0.60–0.85 of that characteristic of the base metal. In these cases, the decrease in the mechanical properties is associated with a sharp change in the microstructure and phase composition of the weld metal, caused by redistribution of alloying elements in the solid solution upon melting and subsequent solidification. In particular, copper precipitates at the grain boundaries.

(3)    Provide the surface formation of laser welded joints.

  • Response. It has been added to the manuscript: ‘The laser welding energy parameters were optimized according to the criterion of the absence of external discontinuities, such as cracks, incomplete fusion zones, undercuts, shrinkage voids, surface pores, etc.

(4)    Please rewritten the conclusions.

  • Response. The conclusions have been rewritten.

(5)    The language should be carefully polished since there are several grammar errors and typos.

  • Response. The manuscript has been double-checked; some typos and mistakes have been corrected.

Reviewer 3 Report

This study investigates the tensile strength characteristics of a laser welded joint of an Al-Cu-Li alloy before and after post-weld heat treatment (PWHT) that included quenching and subsequent artificial aging. The title of the article is very attractive for readers and is practical. The article is well organized, but the following points should be considered.

What is the purpose of research? What is the justification for using the welding process to achieve lower mechanical properties than the base metal?

Choose more appropriate keywords. Some of them (scanning electron microscopy, x-ray diffractometry, mechanical properties) are not mentioned in the abstract.

The abstract must be completely changed and rewritten. Innovation, conducted tests, achievements, and quantitative results are not presented. The purpose of research and innovation should be clearly stated. Also, the performed tests should be presented first, and then the results should be presented quantitatively and qualitatively.

The text of the article needs basic writing and grammar corrections.

Referencing the articles is disappointing (Page , Lines 35, 39, and 44). The use of general sentences with more than four references can be seen in the first paragraph of the introduction. On the other hand, appropriate references were not used to analyze the results.

The introduction is very general. Although the introduction is long, it is written superficially in some paragraphs. Also, in the end, a suitable summary of the importance of the present issue should be provided.

Use the following resources to deepen the introduction. Cooperative effect of strength and ductility processed by thermomechanical treatment for Cu–Al–Ni alloy. Effect of undercooling on microstructure evolution of Cu based alloys. Effects of post-weld heat treatment on the microstructure and mechanical properties of laser-welded NiTi/304SS joint with Ni filler.

On what basis are the welding parameters selected? How is the welding quality confirmed and checked? How has the reproducibility of these results been checked?

Line 176 is dumb; please clarify. Add the error bar to the results. In the conclusion section, a summary of the purpose of the research, innovation, and research method should be presented before presenting the highlights.

No comment.

Author Response

This study investigates the tensile strength characteristics of a laser welded joint of an Al-Cu-Li alloy before and after post-weld heat treatment (PWHT) that included quenching and subsequent artificial aging. The title of the article is very attractive for readers and is practical. The article is well organized, but the following points should be considered.

  • Response.The authors are very grateful to the Reviewer for the spent time and participation. All changes are highlighted in blue

What is the purpose of research? What is the justification for using the welding process to achieve lower mechanical properties than the base metal?

  • Response. This study is devoted to changes in the microstructure and phase composition of the weld metal of the Al-3.9Cu-0.3Mg-1.2Li alloy through its post-weld heat treatment (PWHT). The PWHT procedure has included quenching and subsequent artificial aging, affecting the mechanical properties of the joint. The principal feature of the study is that PWHT has been carried out for welded samples, which are significantly inhomogeneous in terms of their microstructure and phase compositions, as well as the mechanical properties. In such cases, conventional HT procedures, used for manufacturing rolled Al-Cu-Li alloys, are not applicable. So, it has been necessary to optimize PWHT modes, since the phase composition of the base metal should not be changed but hardening phases should be formed in the weld metal. The presented results provide a comprehensive picture of the changes in the mechanical properties of the joint of the V-1469 alloy, occurring upon laser welding and PWHT (quenching and subsequent artificial aging). The solid solution parameters of the weld metal (micro- and nanostructure, the distribution of the alloying elements, the formation and dissolution of the main hardening phases) determines the variations of the mechanical properties.

Choose more appropriate keywords. Some of them (scanning electron microscopy, x-ray diffractometry, mechanical properties) are not mentioned in the abstract.

  • Response. New keywords have been added.

The abstract must be completely changed and rewritten. Innovation, conducted tests, achievements, and quantitative results are not presented. The purpose of research and innovation should be clearly stated. Also, the performed tests should be presented first, and then the results should be presented quantitatively and qualitatively.

  • Response. The abstract has been completely rewritten.

The text of the article needs basic writing and grammar corrections.

  • Response. The manuscript has been double-checked; some typos and mistakes have been corrected.

Referencing the articles is disappointing (Page , Lines 35, 39, and 44). The use of general sentences with more than four references can be seen in the first paragraph of the introduction. On the other hand, appropriate references were not used to analyze the results.

  • Response. These references have been changed to more informative, which reflect the latest trends in the development of aluminum-lithium alloys.

The introduction is very general. Although the introduction is long, it is written superficially in some paragraphs. Also, in the end, a suitable summary of the importance of the present issue should be provided.

  • Response. The introduction has been changed considering the reviewer’s comment.

Use the following resources to deepen the introduction. Cooperative effect of strength and ductility processed by thermomechanical treatment for Cu–Al–Ni alloy. Effect of undercooling on microstructure evolution of Cu based alloys. Effects of post-weld heat treatment on the microstructure and mechanical properties of laser-welded NiTi/304SS joint with Ni filler.

  • Response. The introduction has been changed considering the reviewer’s comment.

On what basis are the welding parameters selected? How is the welding quality confirmed and checked? How has the reproducibility of these results been checked?

  • Response. It has been added to the manuscript: ‘The laser welding energy parameters were optimized according to the criterion of the absence of external discontinuities, such as cracks, incomplete fusion zones, undercuts, shrinkage voids, surface pores, etc. … For each mode, at least three samples were tested. The mean dispersions of the determined values were 2.2% for the ultimate tensile strength, 3.2% for the yield point, and 10% for elongation.

Line 176 is dumb; please clarify. Add the error bar to the results. In the conclusion section, a summary of the purpose of the research, innovation, and research method should be presented before presenting the highlights.

  • Response. Line 176 included technical information; it has been removed. Measurement errors have been added to the text. The conclusions have been rewritten.

Round 2

Reviewer 1 Report

I don't think the authors have responded to the reviewer's comments 1-4 sincerely and satisfactory.

Author Response

Comments:

I don't think the authors have responded to the reviewer's comments 1-4 sincerely and satisfactory.

Response. The authors thank the reviewer for the comment. All changes in the text are highlighted in yellow

Addendum to first comment.

The scheme in fig. 1 is brought into line with the text of the article.

 Addendum to second comment

It has been added to the manuscript photographs of the HAZ before and after PWHT in fig. 8.9. Added information about microstructure changes.

 Addition to the third comment.

We have requested information from All-Russian Scientific Research Institute of Aviation Materials (VIAM). In this study, an Al-3.9Cu-0.3Mg-1.2Li alloy (supplied by VIAM) with original precipitation strengthening after aging treatment was used as the welding base metal.  Added in text

Addition to the fourth comment.

The manuscript includes an optimization of artificial aging depending on various temperature and  time, after quenching at a temperature of 530 0C. The artificial aging regime with maximum mechanical properties was used for a quenching temperature of 560 °C.

Reviewer 2 Report

This paper investigated the laser welding of Al-Cu-Li alloys with quenching and subsequent artificial aging. This paper is well-structured and interesting. Some comments need to be addressed before publication in Metals, as follows:

(1)    For Al-Cu-Li alloys, another effective welding method is friction stir welding with high quality joints. In introduction, the authors should add some discussion about FSW with referenced to https://doi.org/10.1016/j.pmatsci.2020.100706, https://doi.org/10.1007/s40195-022-01444-0

(2)    The primary difficulties of Al-Cu-Li alloys should be stated and discussed.

(3)    Provide the surface formation of laser welded joints.

(4)    Please rewritten the conclusions.

(5)    The language should be carefully polished since there are several grammar errors and typos.

This paper investigated the laser welding of Al-Cu-Li alloys with quenching and subsequent artificial aging. This paper is well-structured and interesting. Some comments need to be addressed before publication in Metals, as follows:

(1)    For Al-Cu-Li alloys, another effective welding method is friction stir welding with high quality joints. In introduction, the authors should add some discussion about FSW with referenced to https://doi.org/10.1016/j.pmatsci.2020.100706, https://doi.org/10.1007/s40195-022-01444-0

(2)    The primary difficulties of Al-Cu-Li alloys should be stated and discussed.

(3)    Provide the surface formation of laser welded joints.

(4)    Please rewritten the conclusions.

(5)    The language should be carefully polished since there are several grammar errors and typos.

Author Response

This paper investigated the laser welding of Al-Cu-Li alloys with quenching and subsequent artificial aging. This paper is well-structured and interesting. Some comments need to be addressed before publication in Metals, as follows:

The authors are very grateful to the Reviewer for the spent time and participation.

All changes in the text are highlighted in yellow

(1)    For Al-Cu-Li alloys, another effective welding method is friction stir welding with high quality joints. In introduction, the authors should add some discussion about FSW with referenced to https://doi.org/10.1016/j.pmatsci.2020.100706, https://doi.org/10.1007/s40195-022-01444-0

Response.  It has been added to the manuscript: ‘It should also be noted another effective method of joining Al-Cu-Li alloys, namely friction stir welding [https://doi.org/10.1016/j.pmatsci.2020.100706, https://doi.org/10.1007/s40195-022-01444-0].

(2)    The primary difficulties of Al-Cu-Li alloys should be stated and discussed.

Response.  It has been added to the manuscript: ‘After laser welding, the ultimate tensile strength of the joints is about 0.60–0.85 of that characteristic of the base metal. In these cases, the decrease in the mechanical properties is associated with a sharp change in the microstructure and phase composition of the weld metal, caused by redistribution of alloying elements in the solid solution upon melting and subsequent solidification. In particular, copper precipitates at the grain boundaries.

(3)    Provide the surface formation of laser welded joints.

Response. It has been added to the manuscript: ‘The laser welding energy parameters were optimized according to the criterion of the absence of external discontinuities, such as cracks, incomplete fusion zones, undercuts, shrinkage voids, surface pores, etc.

(4)    Please rewritten the conclusions.

Response. The conclusions have been rewritten.

(5)    The language should be carefully polished since there are several grammar errors and typos.

Response. The manuscript has been double-checked; some typos and mistakes have been corrected.

Round 3

Reviewer 1 Report

This paper reports the effect of PWHT on transverse tensile properties of laser welded Al-Cu-Li alloy, in relation to the dissolution-precipitation sequence during PWHT. Although it may provide some practically useful information to limited extent, but the results and discussions are not much different from classical textbook knowledge on precipitation-hardenable Al alloys. The authors’ responses are unclear, unconvincing and unsatisfactory to me.

Comments;

1.     “The X-ray diffractograms of the base metal and weld metal” in the caption of Fig.10 is still not strictly correct, because Fig. 10 contains the HAZ result.

2.     The reference [10] is a “Review on failure behaviors of fusion welded high-strength Al alloys due to fine equiaxed zone”. If “The welded joint fractured through the interface between the weld metal and the HAZ (FQZ), as reported by other authors [10]. However, PWHT improved all studied mechanical properties.” on lines 376-378, as similarly as Ref. [10], the changes in microstructures and mechanical properties of FQZ during PWHT should be CLEARLY and EXACTLY shown and discussed.

3.     I don’t understand how the microstructure and mechanical properties of unknown initial state of the base metal can be discussed.

4.     Such a small number of PWHT parameter sets with limited variation in the present manuscript is not sufficient for optimizing the maximum tensile properties of the present joint. If “The authors carried out many trial attempts for optimizing the artificial aging parameters in wide ranges of temperatures and durations”, the summery of the attempts should be shown in a table.

5.     The schematic illustration in Figure 16 is not well supported by experimental data, because size, shape, density and distribution of these precipitates may not be known from the present experiments. Figure 16 should not appear before experimental confirmation.

Author Response

Comments and Suggestions for Authors

This paper reports the effect of PWHT on transverse tensile properties of laser welded Al-Cu-Li alloy, in relation to the dissolution-precipitation sequence during PWHT. Although it may provide some practically useful information to limited extent, but the results and discussions are not much different from classical textbook knowledge on precipitation-hardenable Al alloys. The authors’ responses are unclear, unconvincing and unsatisfactory to me.

Response. 

The authors are very grateful to the reviewer for their time. The sequence of precipitation in Al-Cu-Li alloys is well-established. However, this sequence is poorly studied for welded joints. To establish it, the authors used experimental data obtained with the use of synchrotron radiation, images of micro and nanostructures, as well as classical results and discussions on dispersion-hardening aluminum alloys. A principal feature of the study is that precipitation post-weld heat treatment (PWHT) were carried out on welded samples that were significantly heterogeneous in terms of microstructure, phase composition, and mechanical properties. In such cases, the standard precipitation PWHT procedures used for manufacturing Al-Cu-Li alloys are not applicable. Thus, there is a need to optimize the precipitation PWHT conditions, as the phase composition of the base metal should not change, while reinforcing phases should form in the weld metal.

Comments;

  1. “The X-ray diffractograms of the base metal and weld metal” in the caption of Fig.10 is still not strictly correct, because Fig. 10 contains the HAZ result.

Response.  Figure caption changed. All changes in the text are highlighted in red.

  1. The reference [10] is a “Review on failure behaviors of fusion welded high-strength Al alloys due to fine equiaxed zone”. If “The welded joint fractured through the interface between the weld metal and the HAZ (FQZ), as reported by other authors [10]. However, PWHT improved all studied mechanical properties.” on lines 376-378, as similarly as Ref. [10], the changes in microstructures and mechanical properties of FQZ during PWHT should be CLEARLY and EXACTLY shown and discussed.

Response. The authors, on the basis of reference 10, clearly and exactly discussed the results of changing the microstructure and mechanical properties of FQZ during PWHT. Information added in text.

  1. I don’t understand how the microstructure and mechanical properties of unknown initial state of the base metal can be discussed.

The studies used high-strength corrosion-resistant alloy V-1469T1 (Al-Cu-Mg-Li system) [doi:10.1007/s11015-021-01134-9]. Added in text

  1. Such a small number of PWHT parameter sets with limited variation in the present manuscript is not sufficient for optimizing the maximum tensile properties of the present joint. If “The authors carried out many trial attempts for optimizing the artificial aging parameters in wide ranges of temperatures and durations”, the summery of the attempts should be shown in a table.

Response. Added table with modes

  1. The schematic illustration in Figure 16 is not well supported by experimental data, because size, shape, density and distribution of these precipitates may not be known from the present experiments. Figure 16 should not appear before experimental confirmation.

Response. The authors agree that the schematic illustration is not fully supported by the experimental data. To confirm the size, shape, density and distribution of the hardening phases, it is necessary to conduct studies using transmission electron microscopy. This study is are ongoing and its results will be presented in a next paper.

This figure has been removed.